# Neural Graph Databases

**Maciej Besta**[1,†]    **Patrick Iff**[1]    **Florian Scheidl**[1]    **Kazuki Osawa**[1]    **Nikoli Dryden**[1]
**Michal Podstawski**[2,3]    **Tiancheng Chen**[1]    **Torsten Hoefler**[1,†]

[1]Department of Computer Science, ETH Zurich
[2]Warsaw University of Technology, Warsaw, Poland
[3]TCL Research Europe, Warsaw, Poland

[†]Corresponding authors:
{maciej.besta, torsten.hoefler}@inf.ethz.ch

## Abstract

Graph databases (GDBs) enable processing and analysis of unstructured, complex, rich, and usually vast graph datasets. Despite the large significance of GDBs in both academia and industry, little effort has been made into integrating them with the predictive power of graph neural networks (GNNs). In this work, we show how to seamlessly combine nearly any GNN model with the computational capabilities of GDBs. For this, we observe that the majority of these systems are based on, or support, a graph data model called the Labeled Property Graph (LPG), where vertices and edges can have arbitrarily complex sets of labels and properties. We then develop LPG2vec, an encoder that transforms an arbitrary LPG dataset into a representation that can be directly used with a broad class of GNNs, including convolutional, attentional, message-passing, and even higher-order or spectral models. In our evaluation, we show that the rich information represented as LPG labels and properties is properly preserved by LPG2vec, and it increases the accuracy of predictions regardless of the targeted learning task or the used GNN model, by up to 34% compared to graphs with no LPG labels/properties. In general, LPG2vec enables combining predictive power of the most powerful GNNs with the full scope of information encoded in the LPG model, paving the way for neural graph databases, a class of systems where the vast complexity of maintained data will benefit from modern and future graph machine learning methods.

## 1 Introduction

Graph databases are a class of systems that enable storing, processing, analyzing, and the overall management of large and rich graph datasets [26]. They are heavily used in computational biology and chemistry, medicine, social network analysis, recommendation and online purchase infrastructure, and many others [26]. A plethora of such systems exist, for example Neo4j [100] (a leading industry graph database)[1], TigerGraph [141, 142], JanusGraph [138], Azure Cosmos DB [99], Amazon Neptune [4], Virtuoso [110], ArangoDB [9–11], OrientDB [37, 136], and others [30, 38, 40, 47, 51, 58, 108, 109, 111, 119, 133]. Graph databases differ from other classes of graph-related systems and workloads such as graph streaming frameworks [19] in that they deal with transactional support, persistence, physical/logical data independence, data integrity, consistency, and complex graph data models where both vertices and edges may be of different classes and may be associated with arbitrary properties.

An established data model used in the majority of graph databases is called the *Labeled Property Graph* (LPG) [26]. It is the model of choice for the leading industry Neo4j graph database system. LPG has several advantages over other graph data models, such as heterogeneous graphs [153, 160, 163] or the Resource Description Framework (RDF) [87] graphs, often referred to as knowledge

---

[1]According to the DB engines ranking (https://db-engines.com/en/ranking/graph+dbms)

, Neural Graph Databases. *Proceedings of the First Learning on Graphs Conference (LoG 2022)*, PMLR 198, Virtual Event, December 9–12, 2022.

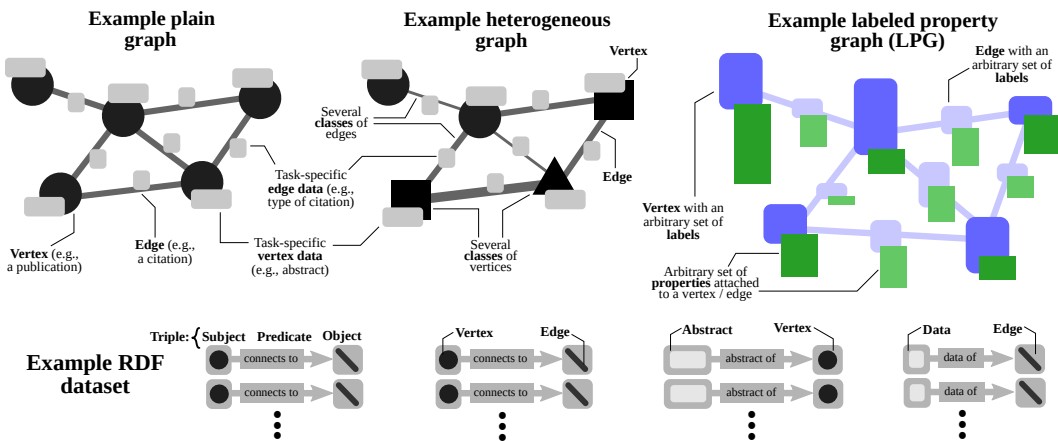

**Figure 1:** Overview of the Labeled Property Graph (LPG) data model used in graph databases, vs. plain and heterogeneous graphs used in broad graph processing and graph machine learning, and RDF triples.

graphs (see Figure 1). First, while heterogeneous graphs support different classes of vertices and edges, LPGs offer *arbitrary sets of labels as well as key-value property pairs* that can be attached to vertices and edges. This facilitates modeling very rich and highly complex data. For example, when modeling publications with graph vertices, one can use labels to model an arbitrarily complex hierarchy of types of publications (journal, conference, workshop papers; best papers, best student papers, best paper runner-ups, etc.). We discuss this example further in Section 2. Second, LPG explicitly stores the neighborhood structure of the graph, very often in the form of adjacency lists [26]. Hence, it enables very fast accesses to vertex neighborhoods and consequently fast and scalable graph algorithms and graph queries. This may be more difficult to achieve in data representations such as sets of triples. First, *any possible relation between any two entities in a graph* (i.e., edges, vertices, and *any* other data) is explicitly maintained as a separate triple (see Figure 1). Second, *any entity is fundamentally the same "resource"*, where "vertex" or "edge" are just roles assigned to a given resource; these roles can differ in different triples (i.e., one resource can be both a vertex and an edge, depending on a specific triple). Hence, RDF graphs may need more storage, and they may require more complex indexing structures for vertex neighborhoods, than in the corresponding LPGs.

Graph neural networks (GNNs) have recently become an established part of the machine learning (ML) landscape [20, 23, 35, 41, 62, 65, 66, 129, 159, 170, 172]. Example applications are node, link, or graph classification or regression in social sciences, bioinformatics, chemistry, medicine, cybersecurity, linguistics, transportation, and others. GNNs have been successfully used to provide cost-effective and fast placement of chips [101], improve the accuracy of protein folding prediction [75], simulate complex physics [114, 127], or guide mathematical discoveries [48]. The versatility of GNNs brings a promise of enhanced analytics capabilities in the graph database landscape.

Recently, Neo4j Inc., Amazon, and others have started to investigate harnessing graph ML capabilities into their graph database architectures. However, current efforts only enable limited learning functionalities that do not take advantage of the full richness of data enabled by LPG. For example, Neo4j's Graph Data Science module [71] supports obtaining embeddings and using them for node or graph classification. However, these embeddings are based on the graph structure, with limited support for taking advantage of the full scope of information provided by LPG labels and properties.

Combining LPG-based graph databases with GNNs could facilitate reaching new frontiers in analyzing complex unstructured datasets, and it could also illustrate the potential of GNNs for broad industry. In this work, we first broadly investigate both the graph database setting and GNNs to find the best approach for combining these two. As a result, we develop **LPG2vec**, an encoder that enables harnessing the predictive power of GNNs for LPG graph databases. In LPG2vec, we treat labels and properties attached to a vertex $v$ as an additional source of information that should be integrated with $v$'s input feature vectors. For this, we show how to encode different forms of data provided in such labels/properties. This data is transformed into embeddings that can seamlessly be used with different GNN models. LPG2vec is orthogonal to the software design and can also be used with any GNN framework or graph database implementation.

We combine LPG2vec with three established GNN models (GCN [85], GIN [161], and GAT [147]), and we show that the information preserved by LPG2vec consistently enhances the accuracy of graph ML tasks, i.e., classification or regression of nodes and edges, by up to 34%. Moreover, LPG2vec supports the completion of missing labels and properties in often noisy LPGs. Overall, it enables **Neural Graph Databases**: the first learning architecture that enables harnessing both the structure and rich data (labels, properties) of LPG for highly accurate predictions in graph databases.

## 2 Background

We first introduce fundamental concepts and notation for the LPG model and GNNs.

### 2.1 Labeled Property Graph Data Model

Labeled Property Graph Model (LPG) [26] (also called the property graph [5]) is a primary established data model used in graph databases. We focus on LPG because it is supported by the majority of systems, and is a model of choice in many leading ones [26] (see Section 1).

At its core, LPG is based on the plain graph model $G = (V, E)$, where $V$ is a set of vertices and $E \subseteq V \times V$ is a set of edges; $|V| = n$ and $|E| = m$. An edge $e = (u, v) \in E$ is a tuple of the out-vertex $u$ (origin) and the in-vertex $v$ (target). If $G$ is undirected, then an edge $e = \{u, v\} \in E$ is a set of $u$ and $v$. $N_i$ and $d_i$ denote the neighbors and the degree of a given vertex $i$ ($N_i \subset V$); $d$ is $G$'s maximum degree. LPG then adds arbitrary *labels* and *properties* to vertices and edges. An LPG is formally modeled as a tuple $(V, E, L, l, K, W, p)$. $L$ is a set of labels. $l : V \cup E \mapsto \mathcal{P}(L)$ is a labeling function, mapping – respectively – each vertex and each edge to a subset of labels, where $\mathcal{P}(L)$ is the power set of $L$, containing all possible subsets of $L$. In addition to labels, each vertex and edge can have arbitrarily many *properties* (sometimes referenced as attributes). A prop-

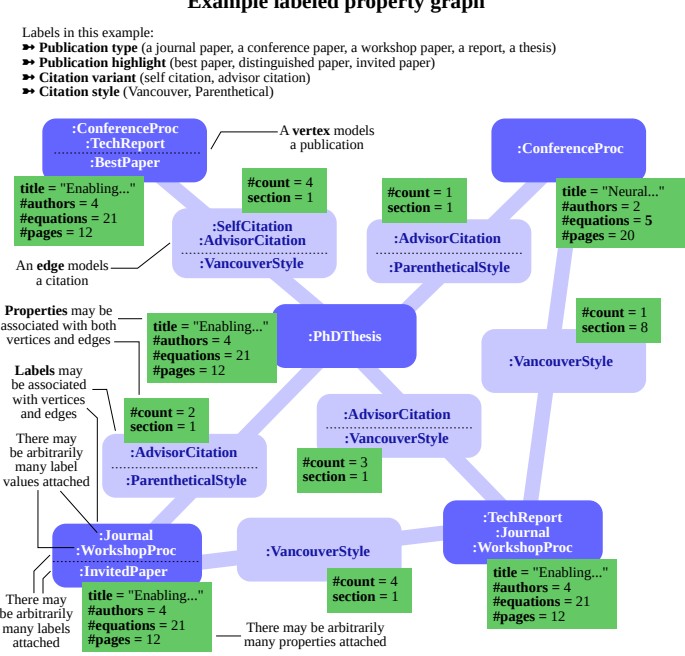

**Example labeled property graph**

Labels in this example:
➡ **Publication type** (a journal paper, a conference paper, a workshop paper, a report, a thesis)
➡ **Publication highlight** (best paper, distinguished paper, invited paper)
➡ **Citation variant** (self citation, advisor citation)
➡ **Citation style** (Vancouver, Parenthetical)

**Figure 2:** An example of an LPG graph modeling publications and citations between them.

erty is a $(key, value)$ pair, with $key$ being an identifier and $value$ being a corresponding value. Here, $K$ is a set with all possible keys and $W$ is a set with all possible values. For any property, we have $key \in K$ and $value \in W$. Then, $p : (V \cup E) \times K \mapsto W$ is a mapping function from vertices/edges to property values. Specifically, $p(u, key)$ and $p(e, key)$ assign – respectively – a value to a property indexed with a key $key$, of a vertex $u$ of an edge $e$. Note that one can assign multiple properties with the same key to vertices and edges (i.e., only the pair $(key, value)$ must be unique).

We illustrate an example of an LPG graph in Figure 2.

### 2.2 Graph Neural Networks

Graph neural networks (GNNs) are a class of neural networks that enable learning over irregular graph datasets [129]. Each vertex (and often each edge) of the input graph usually comes with an *input feature vector* that encodes the semantics of a given task. For example, when vertices and edges model publications and citations between these papers, then a vertex input feature vector

is a encoding of the publication abstract (e.g., a one-hot bag-of-words encoding specifying which words are present). Input feature vectors are transformed in a series of GNN layers. In this process, intermediate hidden latent vectors are created. The last GNN layer produces output feature vectors, which are then used for the *downstream ML tasks* such as node classification or graph classification.

Many GNN models exist [23, 39, 42, 128, 139, 157, 159, 168, 170, 172]. Most of such models consist of a series of GNN layers, and a single layer has two stages: (1) the aggregation stage that – for each vertex – combines the features of the neighbors of that vertex, and (2) the neural stage that combines the results of the aggregation with the vertex score from the previous layer into a new score. One may also explicitly distinguish stage (3), a non-linear activation over feature vectors (e.g., ReLU [85]) and/or normalization. We illustrate a simplified view of a GNN layer in Figure 3.

The input, output, and hidden feature vector of a vertex $i$ are denoted with, respectively, $\mathbf{x}_i, \mathbf{y}_i, \mathbf{h}_i$. We have $\mathbf{x}_i \in \mathbb{R}^k$ and $\mathbf{y}_i, \mathbf{h}_i \in \mathbb{R}^{O(k)}$, $k$ is the dimensionality of vertex input feature vectors. "$(l)$" denotes the $l$-th GNN layer; $\mathbf{h}_i^{(l)}$ are latent features in layer $l$.

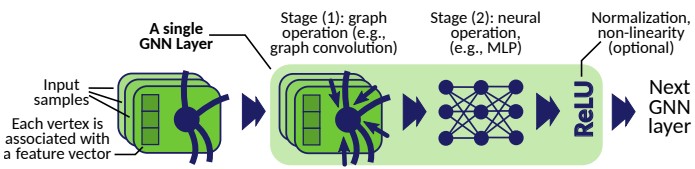

**Figure 3:** Overview of a single GNN layer.

Formally, the graph aggregation stage of a GNN layer can be described using two functions, $\psi$ and $\bigoplus$. First, the feature vector of each neighbor of $i$ is transformed by a function $\psi$. Then, the resulting neighbor feature vectors are aggregated using a function $\bigoplus$, such as sum or max. The outcome of $\bigoplus$ is then processed using a third function, $\varphi$, that models the neural operation and non-linearity. This gives the latent feature vector $\mathbf{h}_i$ in the next GNN layer. Combined, we have

$$\mathbf{h}_i^{(l+1)} = \varphi \left( \mathbf{h}_i^{(l)}, \bigoplus_{j \in N(i)} \psi \left( \mathbf{h}_i^{(l)}, \mathbf{h}_j^{(l)} \right) \right) \tag{1}$$

This is a generic form of GNNs, which can be used to define three major classes of GNNs [34]: *convolutional* GNNs (C-GNNs; examples are GCN [85], GraphSAGE [67], GIN [161], and CommNet [134]), *attentional* GNNs (A-GNNs; examples are MoNet [103], GAT [147], and AGNN [139]), and the most generic *message-passing* GNNs (MP-GNNs; examples are G-GCN [33], EdgeConv [155], MPNN [64], and GraphNets [13]).

To avoid confusion, we always use a term "label" to denote an LPG label, while a term "class" indicates a prediction target in classification tasks.

## 3 Marrying Graph Databases and Graph Neural Networks

We first investigate how LPG-based graph databases and GNNs can be combined to reach new frontiers of complex graph data analytics.

### 3.1 How to Use GNNs with GDBs?

It is not immediately clear on how to use GNNs in combination with the LPG data model. Specifically, labels and properties of a vertex $v$ are often seen as additional "vertices" attached to $v$ [106, 116]. From this perspective, it would seem natural to use them during the aggregation phase of a GNN computation, together with the neighbors of $v$. Similarly, one could consider incorporating attentional GNNs [34], by attending to individual labels and properties. In general, there can be many different approaches for integrating GNNs and LPGs.

Here, we first extensively investigated both the graph database (GDB) and the GNN settings. The goal was to determine the best approach for using GNNs with GDBs in order to benefit the maximum number of different GDB workloads while ensuring a seamless integration with as many GNN models as possible. We consider all major classes of GDB workloads: online transactional, analytical, and serving processing (respectively, OLTP, OLAP, OLSP) [26], and the fundamental GNN model classes (e.g., C-GNN, A-GNN, MP-GNN) [34], for a total of more than 280 analyzed publications or reports.

Our analysis indicated that the most versatile approach for extracting the information from LPG labels and properties is based on encoding labels and properties directly into the input feature vectors,

and subsequently feeding such vectors into a selected GNN model. First, this approach only requires modifications to the input feature vectors, which makes LPG2vec fully compatible with any C-GNN, A-GNN, or MP-GNN model (and many others). Second, this approach is very similar in its workflow to schemes such as positional encodings: is is based on preprocessing and feeding additional information into input feature vectors. Hence, it is straightforward to integrate into existing GNN infrastructures.

## 3.2 Use Cases and Advantages

The first advantage of combining graph databases with GNNs is enhancing the accuracy of traditional GNN tasks: classification and regression of nodes, edges, and graphs (note that tasks such as clustering or link prediction can be expressed as node/edge classification/regression). This is because LPG labels and properties, when incorporated into input feature vectors, carry additional information. This is similar to how different classes of vertices/edges in heterogeneous graphs enhance prediction tasks [160, 163]. However, the challenge is how to incorporate the full rich set of LPG information, i.e., multiple labels and properties, into the learning workflow, while achieving high accuracy and without exacerbating running times or memory pressure.

GNNs can also be used to deliver novel prediction tasks suited for LPG, namely **label prediction** and **property prediction**. In the former, one is interested in assessing whether a given vertex or edge potentially has a specified label, i.e., whether $label \in l(v)$ or $label \in l(e)$, where $label \in L, v \in V, e \in E$ are – respectively – a label, a vertex, and an edge of interest, and $l$ is a labeling function. In the latter, one analogously asks whether a given vertex or edge potentially has a specified property, and – if yes – what its value is, i.e., whether $property \in p(v)$ or $property \in p(e)$, where $p = (k, w), k \in K, w \in W, v \in V, e \in E$ are – respectively – a property, a vertex, and an edge of interest, and $p$ maps $v$ and $e$ to their corresponding properties.

Here, we observe that predicting new labels can be seamlessly resolved with node/edge classification, with the target learned label being $l$. Similarly, property prediction is effectively node/edge regression, where $w$ is the learned value. Thus, it means that one can easily use existing GNN models for *LPG graph completion* tasks, i.e., finding missing labels or properties in the often noisy datasets.

## 3.3 LPG2vec + GNN: Towards A Neural Graph Database

Our architecture for neural graph databases can be seen as an encoder combined with a selected GNN model. An overview is provided in Figure 4.

In the first step to construct an embedding of an LPG, we apply one-hot encoding for labels and properties of each vertex and edge. For labels, the encoding is $\{0, 1\}^{|L|}$, where "1" indicates that a given $i$th label is attached to a given vertex/edge. For properties, the encoding details depend on the property type: If a property can have *discretely many (C) values*, then we encode it using a plain one-hot vector with $C$ entries. A *continuous scalar* property is normalized to $[0; 1]$ or, alternatively, discretized and encoded as a one-hot vector. Importantly, one must use the same norm or discretization for all property instances for a given property type. A *numerical vector* is standardized and normalized. Finally, for properties that contain a *string of text*, we use Sentence Transformers, based on sentence-BERT [120], to embed such a property. String embeddings are usually much longer than other numerical properties to preserve most information in strings.

After encoding, labels and properties are concatenated into input feature vectors for each vertex and for each edge. Importantly, the concatenation is done after ordering the elements of a set Labels ∪ Property keys (i.e., $L \cup K$) and applying the same ordering for each vertex and for each edge. This ensures that the embeddings of labels/properties follow the same order in each feature vector and that the lengths of feature vectors for, respectively, vertices and edges, are the same.

## 3.4 Seamless Integration with GNN Models and Encodings

Both vertex and edge information is straightforwardly harnessed by LPG2vec by first encoding the input vertex or edge labels/properties within LPG2vec. Then, we feed such vertex and edge encodings, as input feature vectors $\mathbf{x}_i$ (for any vertex $i$) and $\mathbf{x}_{ij}$ (for any edge $(i, j)$), into a selected GNN model. Here, LPG2vec enables seamless integration with virtually any GNN model, encoding, or architecture. This is due to the simplicity of our solution: all LPG2vec does is providing "enriched" input feature vectors $\mathbf{x}_i$ and $\mathbf{x}_{ij}$. Vectors $\mathbf{x}_i$ can be directly fed to any convolutional, attentional, or

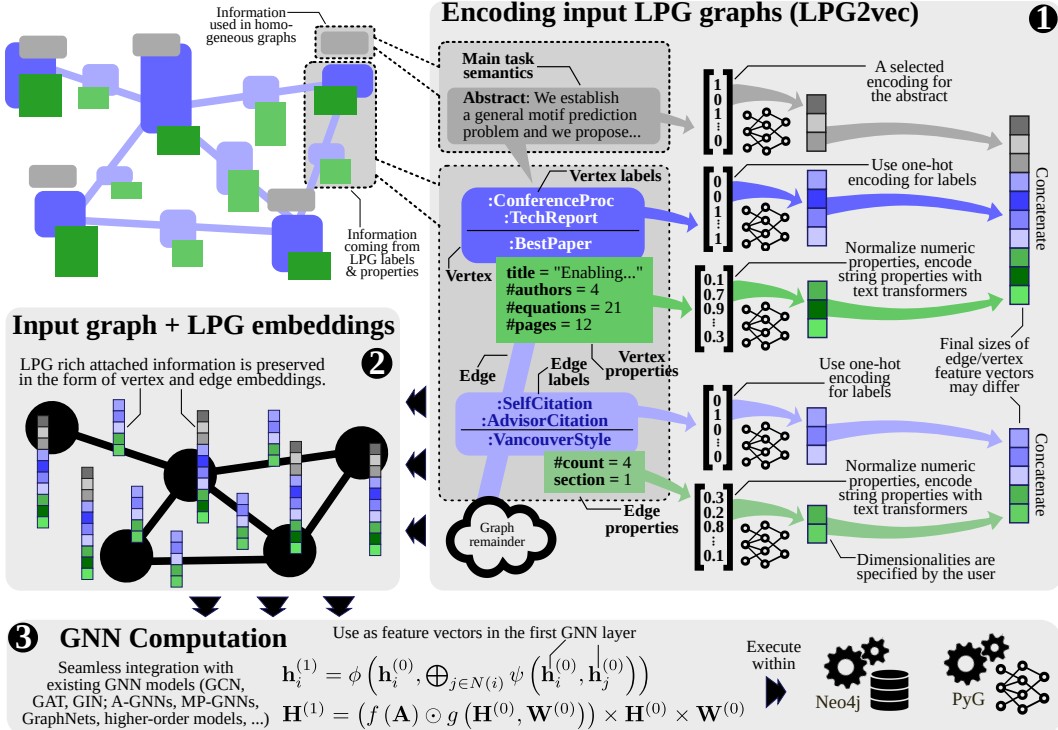

**Figure 4:** An overview of LPG2vec in the context of processing LPG graphs with GNNs. First ("❶"), the graph data is loaded from disk and encoded using LPG2vec. Here, we differentiate the additional data usually used with homogeneous graphs, that determines the task semantics (in this case, publication abstracts), from the LPG-related additional data (labels, properties). Note that, in practice, encoding the abstract could be just implemented as encoding an additional property. The encoding process gives a graph dataset ("❷") that is ready for the actual GNN computation that can be executed in a dedicated module of a graph database (e.g., Neo Graph Data Science) or in a dedicated ML framework (e.g., PyG). Importantly, LPG2vec preserves all the rich LPG information in the form of vertex and edge embeddings. Thus, the actual input to the GNN computation is a homogeneous graph structure together with the embeddings. This makes the integration with existing GNN models straightforward ("❸"). The computation itself is conducted in a dedicated module of a graph database (e.g., Neo4j's Graph Data Science), but - thanks to the seamless LPG2vec design (i.e., the fact that the output of LPG2vec is a homogeneous graph with enhanced feature vectors) - it can also be conducted in a standalone GNN framework (e.g., PyG).

message-passing GNN model, as the input vertex feature vectors. Vectors $\mathbf{x}_{ij}$ are fed into any model that also incorporates edge feature vectors.

Moreover, LPG2vec also enables easy integration with encoding schemes such as MPGNNs-LSPE [53]. This can be achieved by, for example, concatenating the LPG2vec vectors with any additional encodings, and then using the resulting feature vectors with the selected GNN model.

## 4 Evaluation

Our main goal in the evaluation is to show that LPG2vec successfully harnesses the label and property information from the LPG graph datasets to offer more accurate predictions in graph ML tasks. Our analysis comes with a large evaluation space. Thus, we show selected representative results; full data is in the appendix due to space constraints.

### 4.1 Experimental Setup

An important part of the experimental setup is finding the **appropriate graph datasets** that *have many labels and properties*. First, we use the Microsoft Academic Knowledge Graph (MAKG) [56]. The original graph is in the RDF format. We extracted data from RDF triples describing consecutive vertices, and we built LPG vertex entries containing the gathered data; single triples containing edges were parsed directly into the LPG format. Due to the huge size of MAKG, we extracted two subgraphs. For this, we consider the following LPG labels of vertices: *:Paper, :Author, :Affiliation, :ConferenceSeries, :ConferenceInstance, :FieldOfStudy*, as well as the links between them. Then, we additionally limit the number of the considered research areas (and thus vertices) in the *:FieldOfStudy* field (four for a small MAKG dataset, 25 for a large MAKG dataset); they form classes to be predicted.

For diversified analysis, we make sure that these two datasets differ in their degree distributions, implying different connectivity structure. Second, we use example LPG graphs provided by Neo4j[2]; While these datasets are small, they are original excerpts from industry LPG databases. Most importantly, we use a "citations" network (modeling publications and citations between them), a "Twitter trolls" network (modeling anonymized Twitter trolls and the interaction of retweets), and a network modeling crime investigations. The details of datasets are in Table 1; the appendix provides a full specification of the associated labels/properties in selected datasets, as well as additional results.

While there are many heterogeneous graphs available online, they have usually single labels (often called types) per vertex or edge. We considered some of these graphs; we first convert them appropriately into the LPG model by transforming certain information from the graph structure into labels and properties. Note that we do *not* compete with heterogeneous representations, datasets, and the associated heterogeneous GNN models (they are outside the scope of this work); instead, we focus on LPG because this is the main established graph data model in graph databases.

| Dataset | #vertices | #edges | #labels | #properties | size | Prediction target & ML task details |
|---|---|---|---|---|---|---|
| **[MAKG]** (small) | 3.06M | 12.3M | 20 | 28 | 1.2 GB | Publication area (node classification, 4 classes) |
| **[MAKG]** (large) | 50.7M | 190M | 20 | 28 | 19.5 GB | Publication area (node classification, 25 classes) |
| **[Neo4j]** citations | 132k | 221k | 5 | 6 | 51 MB | Citation count (node regression) |
| **[Neo4j]** Twitter trolls | 281k | 493k | 13 | 14 | 79 MB | Retweet count (node regression) |
| **[Neo4j]** crime investigations | 61.5k | 106k | 28 | 29 | 17 MB | Crime type (node classification) |

Table 1: **Considered LPG datasets & ML tasks**. **[Neo4j]**: provided by the Neo4j online repository, **[MAKG]**: extracted from MS Academic Knowledge Graph. Additional results for all the datasets are provided in the appendix.

We consider different established GNN models: GCN [85] (a seminal convolutional GNN model), GAT [147] (a seminal attentional GNN model), and GIN [161] (a seminal model having more expressive power than GCN or GAT). We test these models with and without the LPG2vec encoding scheme. Then, when considering models enhanced with LPG2vec, we test variants that harness the additional LPG information coming from only labels, only properties, and from both labels and properties. Our goal is to investigate how exactly the rich additional LPG information influences the accuracy of the established graph ML tasks, focusing on node classification (assigning each vertex to one of a given number of classes) and node regression (predicting a real value for each vertex) [159].

We split the datasets into train, val, and test by the ratio of $[0.8, 0.1, 0.1]$. We set the mini-batch size to 32, use the Adam optimizer [82], the learning rate of 0.01 augmented with the cosine annealing decay, and we train for 100 epochs. The node mini-batch sampling is conducted using the GraphSAINT established scheme [167]. We use the cross-entropy and MSE loss functions for classification and regression, respectively. In the design of used GNN models (GCN, GAT, GIN), following the established practice [165], we incorporate one preprocessing MLP layer, followed by two actual GNN layers, and then one additional post-processing MLP layer. We use the PReLU non-linearity.

Our implementation is integrated into PyG [57]. We use GraphGym [165] as well as Weights & Biases [29] for managing experiments.

## 4.2 Improving the Accuracy with LPG Labels and Properties

We first analyze how LPG2vec appropriately harnesses the rich information from LPG labels/properties, enabling accuracy improvements for different GNN models. Example results are in Figure 5, showing both node classification and node regression, with MAKG and Neo4j datasets. We plot the the final test accuracy (with the standard deviation) for classification and the mean absolute error (MAE) for node regression. The task is to predict the research area of the publication (for MAKG) and the citation count of a paper (for Neo4j citations). In the results, the baseline with no LPG labels/properties (i.e., only the neighborhood structure) consistently delivers the lowest accuracy (MAKG), or – in some cases such as for the GCN/GAT models and Neo4j citations dataset – is unable to converge. Then, for MAKG, including, respectively, labels (describing *paper types*), a property (*paper title*), and both the labels and the title property, steadily improves the accuracy, reaching nearly 35% for GCN. The trend is similar across all the studied models, and they achieve similarly high accuracy, which indicates that harnessing the appropriate labels/properties is very relevant and - when this information is present - different GNN models will perform similarly well. Neo4j citations

---

[2]Available at https://github.com/neo4j-graph-examples

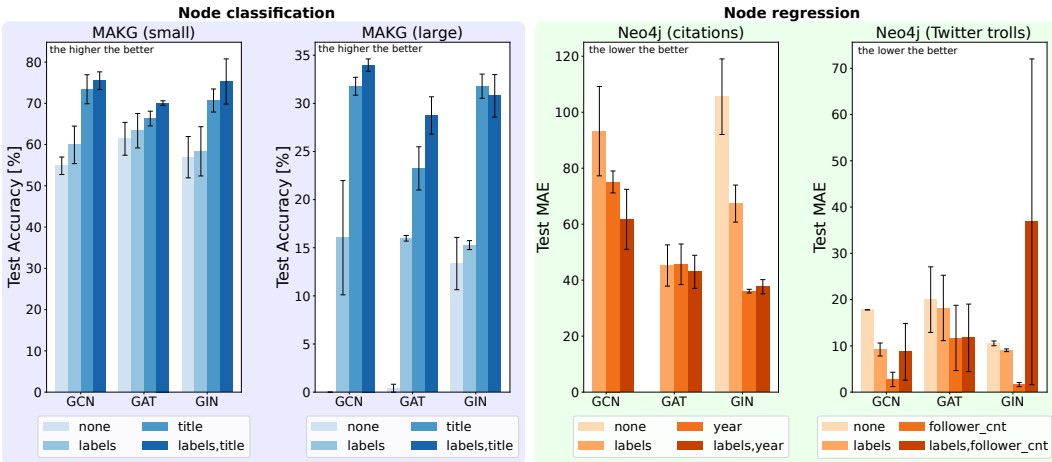

**Figure 5:** Advantages of preserving the information encoded in LPG labels and properties, for node classification (the MAKG datasets in the left panel; 4 classes for small and 25 classes for large) and node regression (the Neo4j datasets in the right panel).

and Twitter trolls are similar (note the different metric as this is a node regression task). The main difference is that, for GIN and the Twitter dataset, combining the labels and the *follower count* leads to worse results than only using one of these two individually. This illustrates that certain combinations of LPG information might not always enhance the accuracy; we study this in more detail in Sec. 4.3. Another interesting effect takes place when considering the bare graph structure on the small MAKG. Here, GCN performs worst, GAT is somewhat better, while GIN delivers much higher accuracy than GAT. We conjecture this is because GIN is provably highly expressive in the Weisfeiler-Lehman sense (when considering the bare graph structure) [161]. Overall, the results show the importance of including both the labels and properties when analyzing LPG graphs.

## 4.3 Selecting the Right Labels and Properties

In some experiments, we observed that selecting certain properties was not improving the accuracy. Moreover, in certain cases, the accuracy was actually diminishing. We analyze this effect in more detail in Figure 6 for the node classification and regression on MAKG small and Neo4j Twitter trolls, with the GIN model, plotting both train and test accuracy. The plots show the impact of using each of the many available properties on the final prediction accuracy. For example, on the small MAKG, using the *title* property significantly improves the accuracy, and the majority of other properties also increase it, although by much smaller (often negligible) factor. Still, using the *publication date* property in many cases decreases the accuracy (see the bottom-left plot). We further analyze this effect with heatmaps, by considering each possible *pair of properties*, and how using this pair impacts the results. The accuracy is almost always enhanced, when using *title* together with nearly any other property. Some properties, such as *entity id*, have no effect. Many pair combinations result in slight accuracy improvements. However, in the Neo4j Twitter case (the bottom panel), in the test accuracy, while using many individual properties significantly enhances the accuracy, most combinations of property pairs decrease it. Interestingly, this only happens for the GIN model; *the GCN models and GAT models are able to extract useful knowledge from most property pairs* (these results are provided in the appendix, see Figures 12 and 13). This illustrates that it is important to understand the data and select the right encoded LPG information *and* the model for a given selected graph ML task.

## 5 Related Work and Discussion

Our work touches on many areas. We now briefly discuss related works. We do not compare LPG2vec to non-GNN baselines because our main goal is to illustrate how to integrate GNN capabilities into GDBs, and *not* to argue that neural methods outperform those of traditional non-GNN baselines. Hence, we do not focus on experiments with traditional GDB non-neural tasks such as BFS or Connected Components.

**Graph Neural Networks and Graph Machine Learning** Graph neural networks (GNNs) emerged as a highly successful part of the graph machine learning field [66]. Numerous GNN models have been developed [20, 23, 35, 41, 62, 66, 129, 159, 170, 172], including convolutional [67, 85, 134, 156, 161],

**MAKG small, node classification (the higher the better), GIN model**

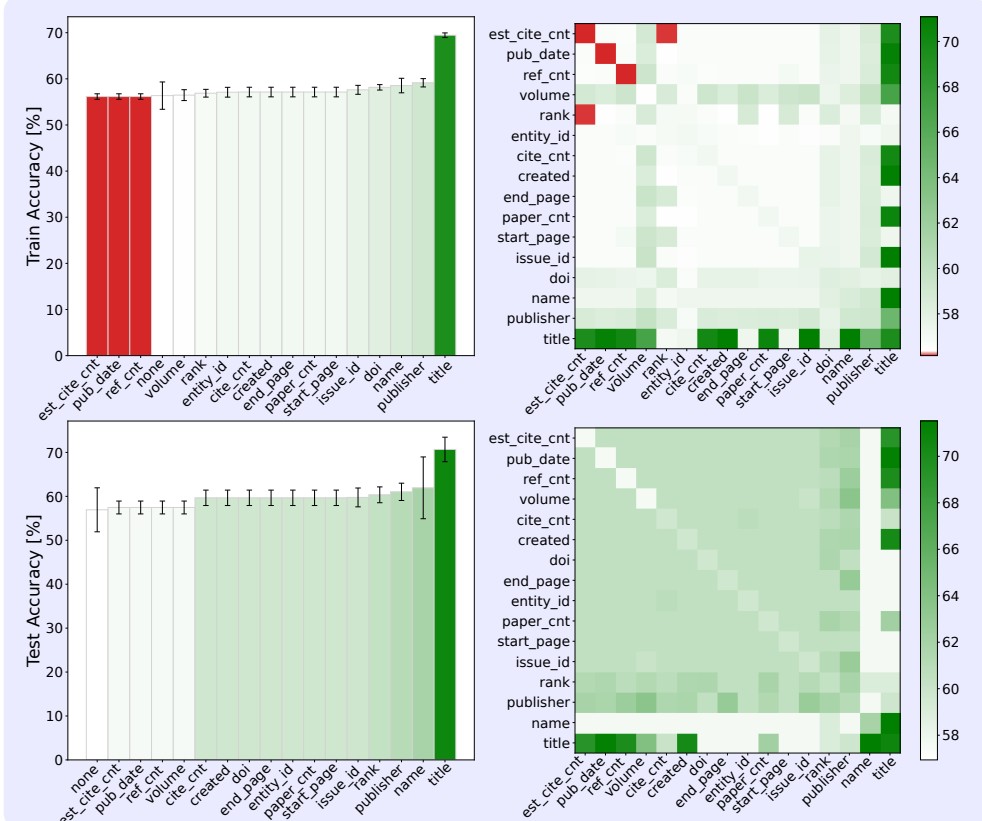

**Neo4j Twitter, node regression (the lower the better), GIN model**

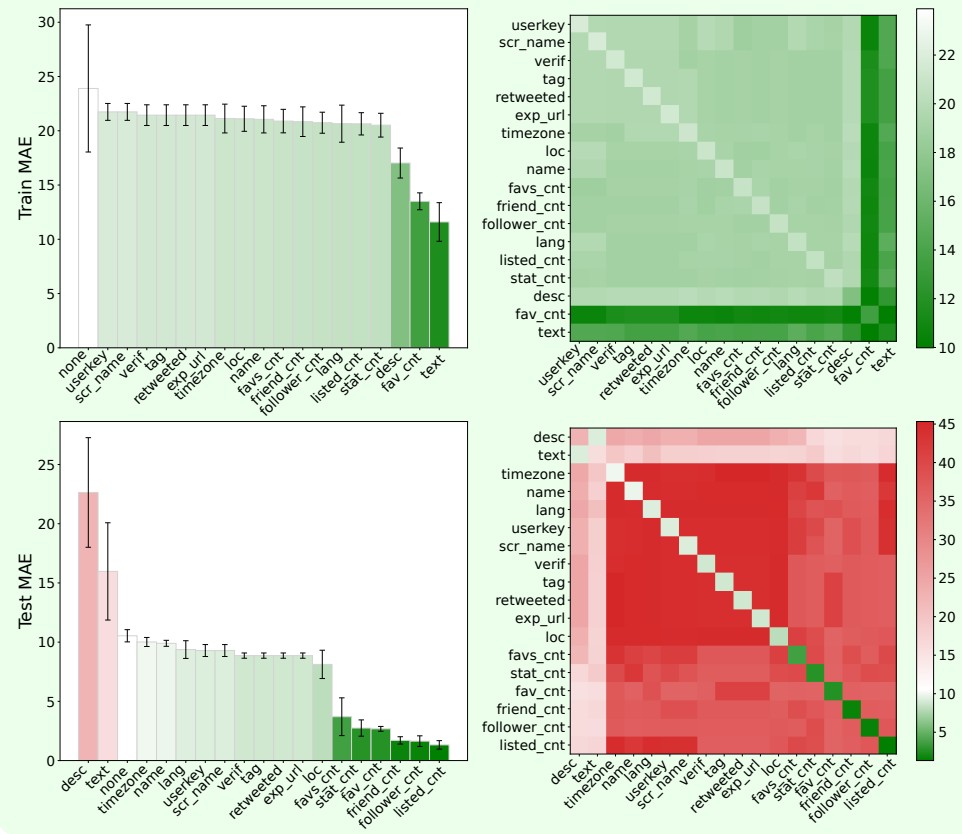

**Figure 6:** Analysis of the impact from different properties on the accuracy, considering both individual properties and the combinations of property pairs, for the MAKG and Neo4j, with the node classification and regression tasks, for the GIN model. The green color indicates that a given result is better than for a graph with no labels/properties. The red color indicates the results are worse than for a graph with no labels/properties.

attentional [103, 139, 147], message-passing [13, 33, 64, 127, 155], or – more recently – higher-order ones [1, 2, 14, 31, 104, 124, 125]. Moreover, a large number of software frameworks [57, 72, 74, 89, 93, 144, 148–151, 154, 158, 169, 171, 173], and even hardware accelerators [60, 83, 84, 90, 162] for processing GNNs have been introduced over the last years. LPG2vec enables using all these designs together with the LPG graphs and consequently with LPG-based graph databases. This is because of the fact that the information within LPG labels and properties is encoded into the input features vectors, which can then be seamlessly used with essentially any GNN model or framework of choice.

**Graph Databases (GDBs)** [26] are systems used to manage, process, analyze, and store vast amounts of rich and complex graph datasets. GDBs have a long history of development and focus in both academia and in the industry, and there has been significant work on them [6, 7, 49, 59, 68, 78, 86]. A lot of research has been dedicated to graph query languages [5, 5, 32], GDB management [32, 76, 100, 113, 115], compression in GDBs and data models [16, 22, 25, 28, 92, 94, 105], execution in novel environments such as the serverless setting [45, 96, 143], and others. Many GDBs exist [4, 8–11, 30, 37, 38, 40, 47, 51, 52, 55, 58, 79, 80, 97–99, 107–111, 118, 119, 121, 133, 136–138, 141, 142, 166, 174, 175]. We enhance the learning capabilities of graph databases by illustrating how to harness all the information encoded in Labeled Property Graph (LPG), a data model underlying the majority of graph databases, and use it for graph ML tasks such as node classification.

**Resource Description Framework and Knowledge Graphs** Resource Description Framework (RDF) [87] is a standard originally developed as a data model for metadata. It consists of triples, i.e., 3-tuples used to encode any information in a graph. Hence, while it has been used to encode knowledge in ontological models and in frameworks called RDF stores [69, 102, 112], it is less common in graph databases that focus on achieving high performance, low latency, and large scale. The reason is that LPG facilitates explicit storage of graph structure, and thus makes it easier to achieve high performance of different graph algorithms and complex business-intelligence graph queries that commonly require accessing graph neighborhoods [26]. We focus on graph databases built on top of LPG, and thus RDF and the associated graph ML models such as RDF2vec [77, 117, 122, 123] are outside the scope of this work. Note that the notions of label and property prediction are analogous to the concepts of knowledge graph completion [3, 12, 43, 46, 91, 91, 130, 131, 135, 145, 146, 152, 164].

**Dynamic, Temporal, and Streaming Graph Processing Frameworks** There also exist systems for processing dynamic, temporal, and streaming graphs [15, 19, 44, 126]. Their setting partially overlaps with graph databases, because they also focus on high-performance graph processing and on solving (in a dynamic setting) graph problems such as Betweenness Centrality [95, 132], Graph Traversals [21, 24, 36, 81], Connected Components, Graph Coloring, Matchings, and many others [17, 18, 27, 50, 54, 61, 63, 70, 73, 88, 140]. However, the rate of updates in such systems is much higher than in graph databases, thus requiring usually significantly different system designs and architectures. More importantly, such frameworks usually do not focus on rich data and do not use the LPG model. Hence, these systems differ fundamentally from graph databases, and are outside the focus of this paper.

## 6 Conclusion

Graph databases (GDBs), despite being an important part of the graph analytics landscape, have still not embraced the full predictive capabilities of graph neural networks (GNNs). To address this, we first observe that the majority of graph databases use, or support, the Labeled Property Graph (LPG) as their data model. In LPG, the graph structure, stored explicitly in the compressed-sparse row format, is combined with labels and key-value properties that can be attached, in any configuration, to vertices and edges. To integrate GDBs with graph machine learning capabilities, we develop LPG2vec, an encoder that converts LPG labels and properties into input vertex and edge embeddings. This enables seamless integration of any GDB with any GNN model of interest.

Our evaluation shows that incorporating labels and properties into GNN models consistently improves accuracy. For example, GCN, GAT, and GIN models achieve even up to 34% better accuracy in node classification for the LPG representation of the Microsoft Academic Knowledge Graph, compared to a setting without LPG labels and properties. We conclude that LPG2vec will facilitate the development of neural graph databases, a learning architecture that harnesses both the structure and rich data (labels, properties) of LPG for highly accurate predictions in graph databases. It will lead to the wider adoption of GNNs in the broad graph database industry setting.

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

# Appendix

## A  Dataset Specification

We present the details about the used datasets.

### A.1  MAKG

The dataset *MAKG* (small) consists of $3'066'782$ vertices and $12'314'398$ edges. Each vertex is labeled with either *author* (55%), *paper* (44%), *affiliation* ($< 1\%$), *conferenceseries* ($< 1\%$), *conferenceinstance* ($< 1\%$), *fieldofstudy* ($< 1\%$), or *journal* ($< 1\%$). Vertices with the label *paper* are further subdivided into *book*, *bookchapter*, *conferencepaper*, *journalpaper*, *patentdocument* or *others*. Vertices labeled with *affiliation*, *author*, *conferenceseries*, *conferenceinstance*, *fieldofstudy* and *journal* do all have the properties *rank*, *name*, *papercount*, *citationcount*, and *created*. Some of them have additional properties, e.g., *homepage*. All vertices with the label *paper* have the properties *rank*, *citationcount*, *created*, *title*, *publicationdate*, *referencecount*, and *estimatedcitationcount*. Some of them have the additional properties *publisher*, *volume*, *issueidentifier*, *startingpage*, *endingpage*, or *doi*. Edges do not have properties but each edge has a label which is either *cites* (40%), *creator* (36%), *hasdiscipline* (11%), *apreasinjournal* (6%), *memberof* (5.7%), *appearsinconferenceinstance* ($< 1\%$), *appearsinconferenceseries* ($< 1\%$), or *ispartof* ($< 1\%$).

### A.2  Citations

The *citations* dataset contains $132'259$ vertices and $221'237$ edges; we use it mostly for debugging purposes. Each vertex has one label which is either *author* (61%), *article* (39%), or *venue* ($< 1\%$). All *article*-vertices have the properties *index* (a 32-digit HEX number), *title* and *year* (the year in which the article was published). 85% of the articles have the property *abstract* and 72% of them have the property *ncitations* (the article's citation count). Vertices labeled with *author* or *venue* have only one property called *name*. Edges do not have properties but each edge has a label which is either *author* (64%), *venue* (23%), or *cited* (13%).

### A.3  Twitter

The dataset *TwitterTrolls* contains $281'136$ vertices and $493'160$ edges. Vertices are labeled with *tweet* (82%), *url* (8%), *hashtag* (5%), *user* (5%), *trolluser* ($< 1\%$), or *source* ($< 1\%$). Vertices wit labels *hashtag*, *source*, *user*, and *url* have a single property each, namely *tag*, *name*, *userkey*, and *expandedurl* respectively. Vertices labeled with *trolluser* do all have the properties *sourcename* and *userkey*. Most of them ($> 80\%$) have additional properties *lang* (language), *verified* (true or false), *name*, *description*, *location*, *timezone*, *createdat*, *favoritescount*, *followerscount*, *friendscount*, *listedcount*, and *statusescount*. Most vertices labeled with *tweet* ($> 80\%$) have properties *createdat*, *createdstr* and *text*. About 25% of them have additional properties *favoritecount*, *retweetcount*, and *retweeted*. Edge do not have properties but each edge has a label which can be *posted* (41%), *hastag* (22%), *postedvia* (12%), *mentions* (11%), *retweeted* (8%), *haslink* (6%), or *inreplyto* ($< 1\%$).

### A.4  Differences to Traditional GNN Datasets

The main difference between LPG graphs and traditional GNN datasets such as Citeseer or Cora is that the latter usually do not have extensive sets of labels. Instead, these datasets often have vertices from different classes, which may be interpreted as a single label (that would encode such

different classes). Moreover, these datasets often do not have rich sets of attached *different* properties. Instead, they may come with extensive feature vectors that encode a single large additional piece of information, for example a whole abstract. Finally, in the graph database setting, it is less common to process graphs such as PROTEINS, where the dataset consists of a very large number of relatively small graph. Instead, it is more common to focus on one large graph dataset.

## B    Results for Additional Labels, Properties, and Datasets

Figures 7–18 illustrate the impact of using each of the many available properties, and pairs of properties, on the final prediction accuracy. We show results separately for each GNN model and also aggregated for all thee models, for the completeness of the analysis. To facilitate comparing the data, we also replot the results for the GIN model for MAKG small and Neo4j Twitter analyses from Section 4. Finally, Figure 19 shows results for an additional Neo4j dataset modeling crime investigations.

Interestingly, the largest accuracy increase for MAKG is consistently obtained when including the title property. This is the case for all the considered GNN models. Similarly, when detecting trolls, including the counts of friends or followers was crucial in consistent accuracy improvements. This indicates that it is more important to appropriately understand the data and include the right information in the input feature vectors, and once this is achieved, different GNN models would be similarly able to extract this information for more accurate outcomes.

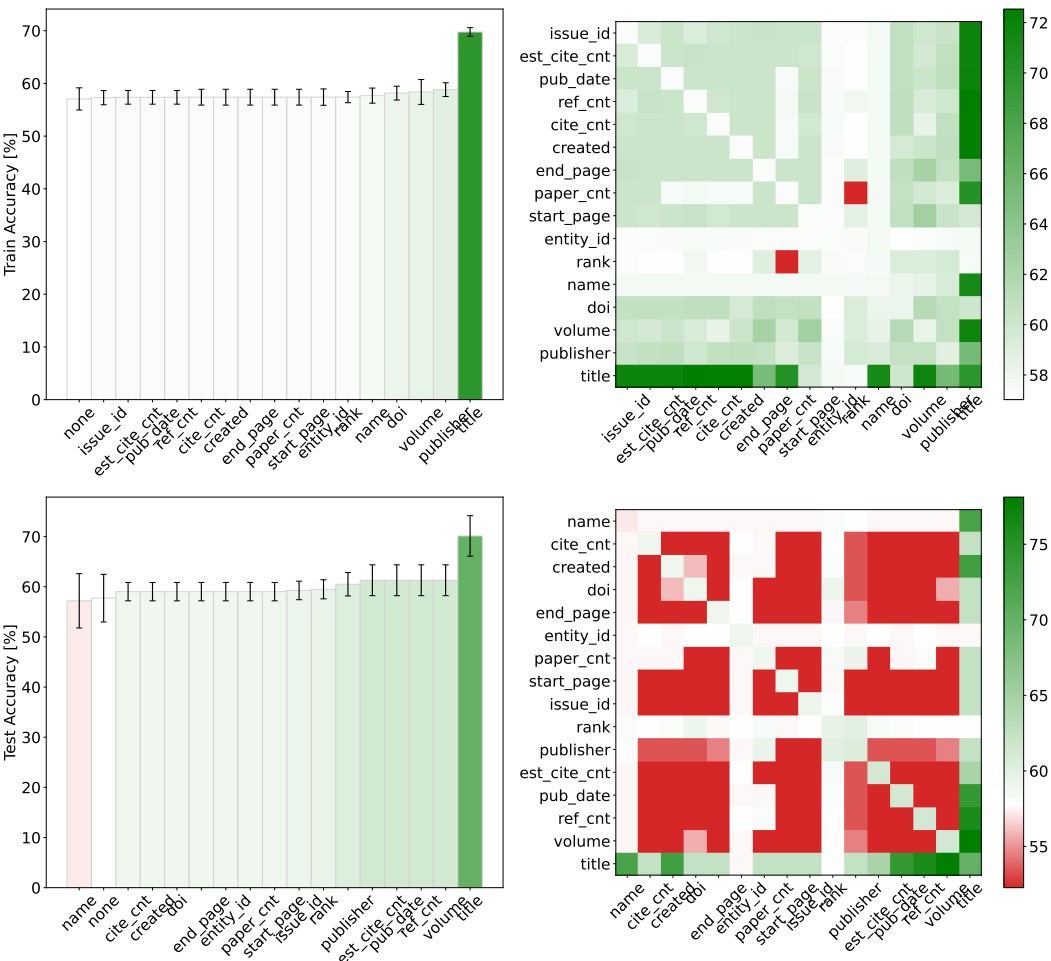

**Figure 7: MAKG small (node classification, 4 classes, results aggregated over all three models)**. Impact from different properties and their combinations on the accuracy. Green: accuracy is better than that of a graph with no labels/properties; red: the accuracy is worse than that of a graph with no labels/properties.

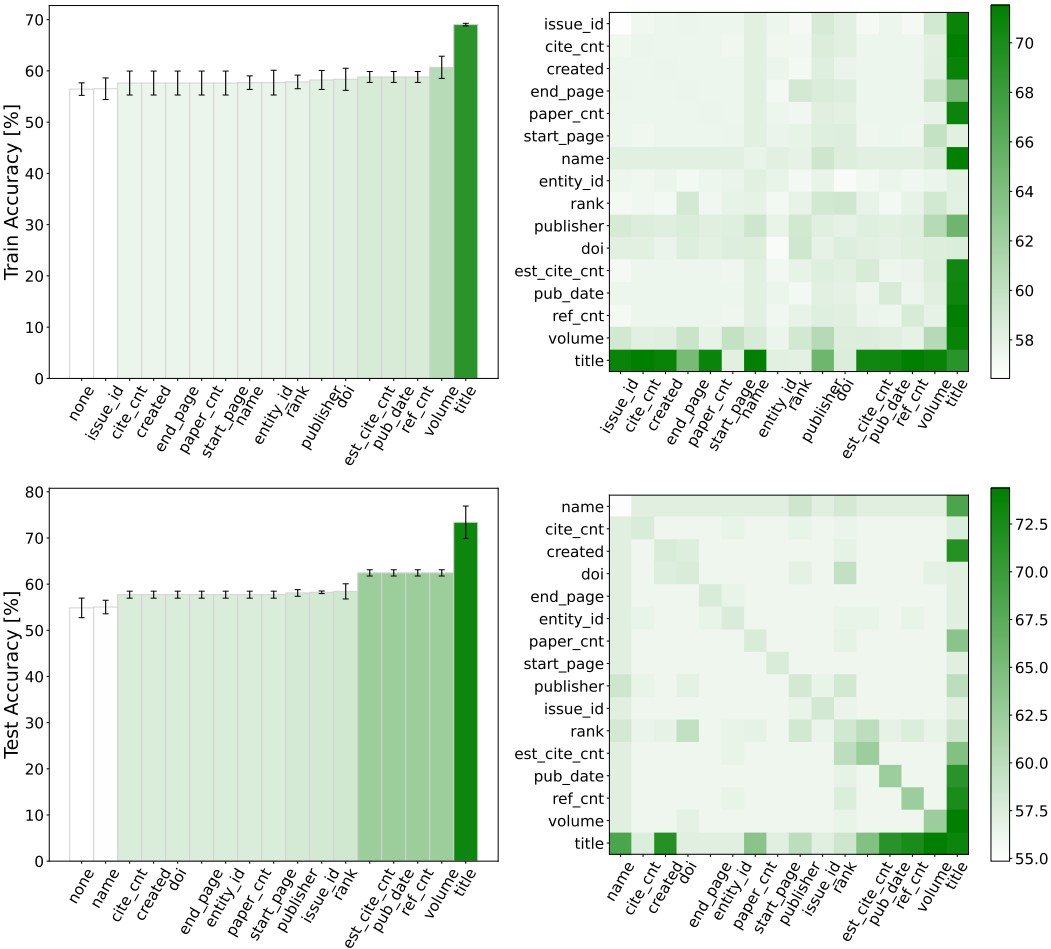

**Figure 8: MAKG small (node classification, 4 classes, GCN-only results).** Impact from different properties and their combinations on the accuracy. Green: the accuracy is better than that of a graph with no labels/properties; red: the accuracy is worse than that of a graph with no labels/properties.

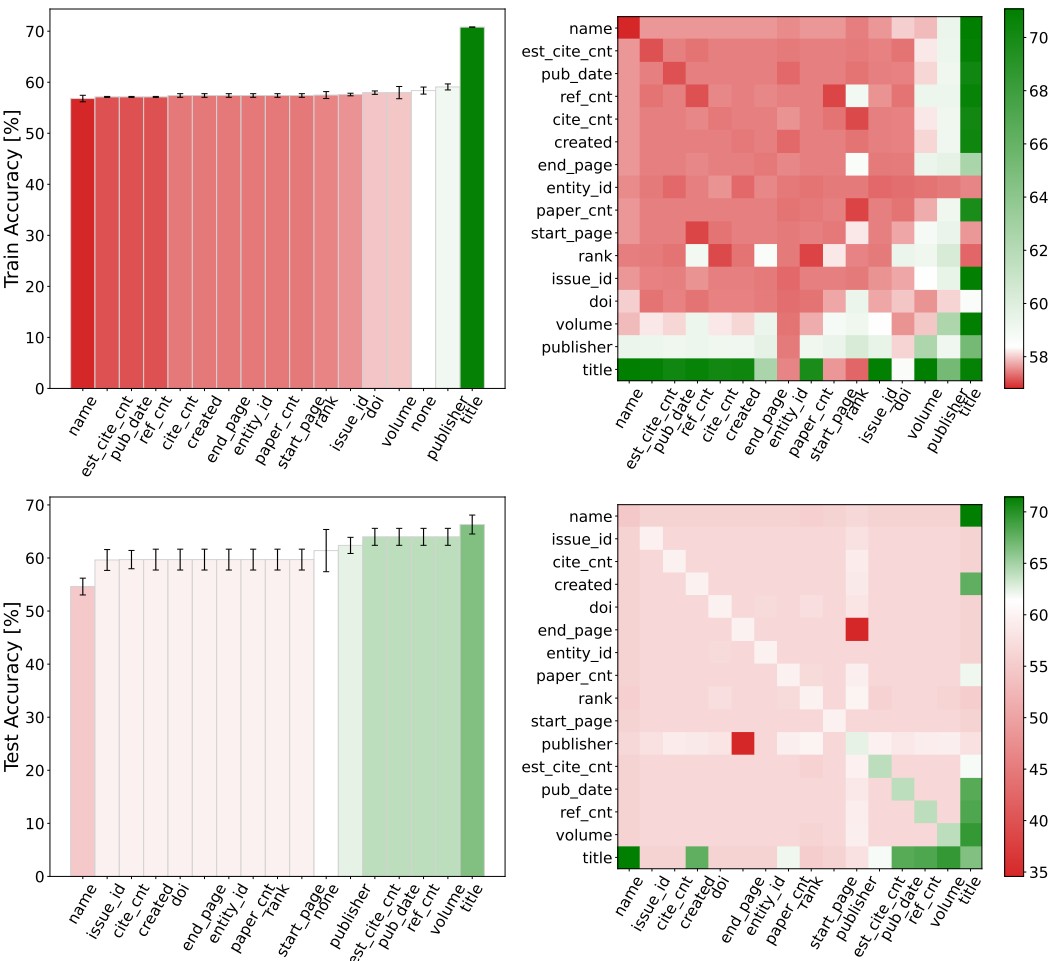

**Figure 9: MAKG small (node classification, 4 classes, GAT-only results)**. Impact from different properties and their combinations on the accuracy. Green: accuracy is better than that of a graph with no labels/properties; red: the accuracy is worse than that of a graph with no labels/properties.

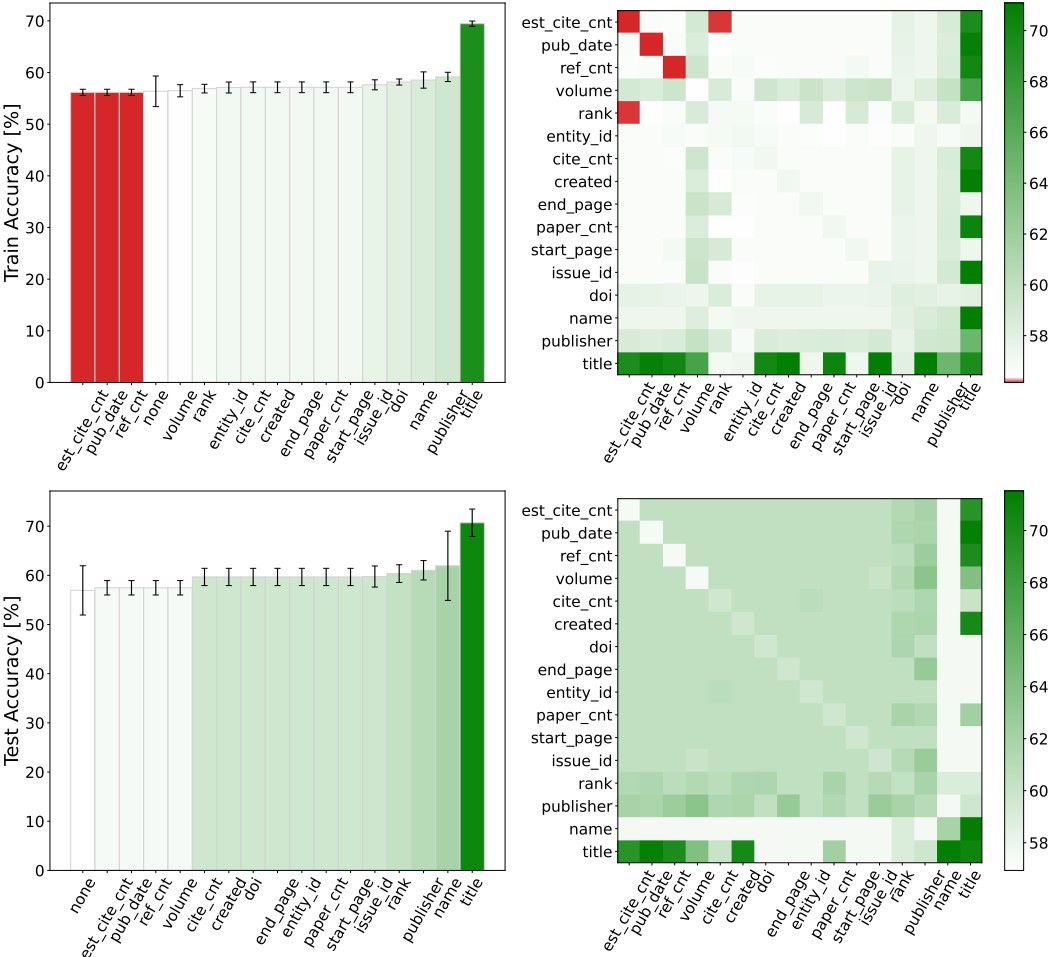

**Figure 10: MAKG small (node classification, 4 classes, GIN-only results)**. Impact from different properties and their combinations on the accuracy. Green: accuracy is better than that of a graph with no labels/properties; red: the accuracy is worse than that of a graph with no labels/properties.

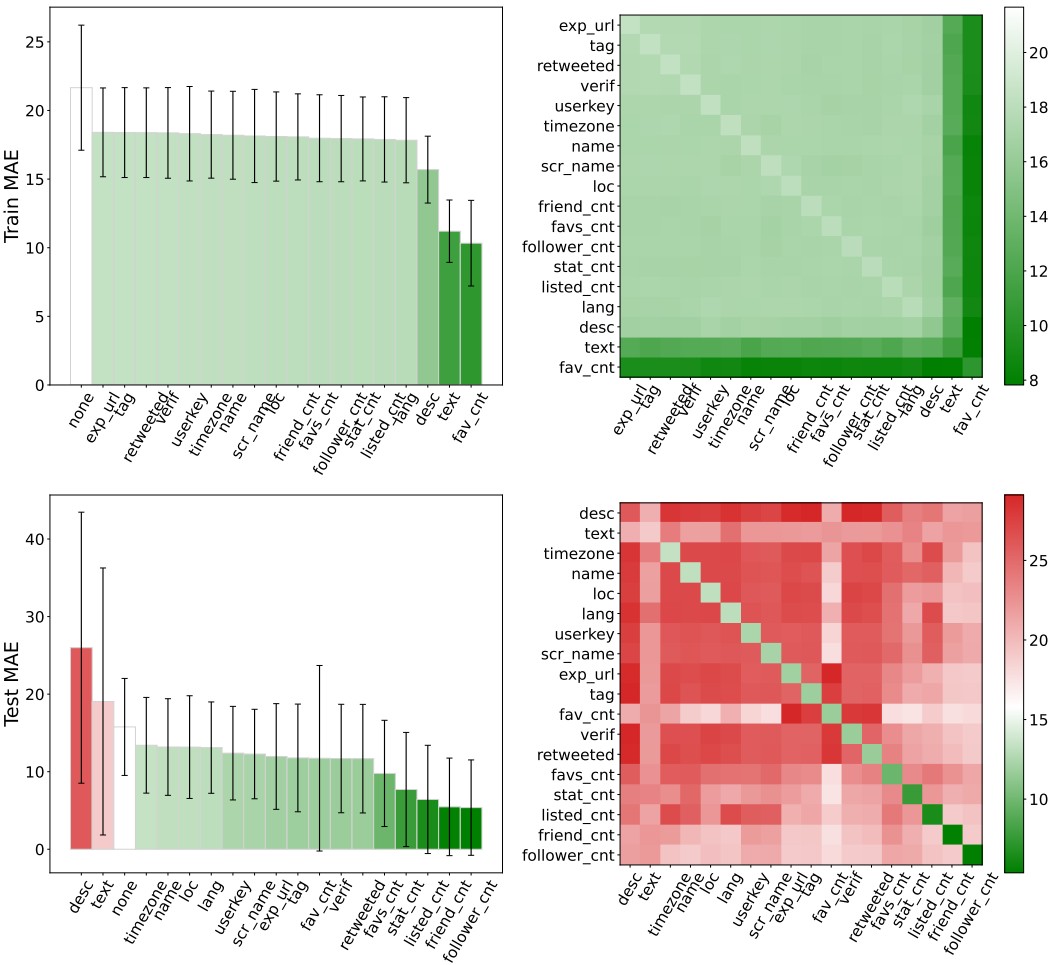

**Figure 11: Neo4j Twitter trolls (node regression, results aggregated over all three models).** Impact from different properties and their combinations on the MAE. Green: MAE is better than that of a graph with no labels/properties; red: the MAE is worse than that of a graph with no labels/properties.

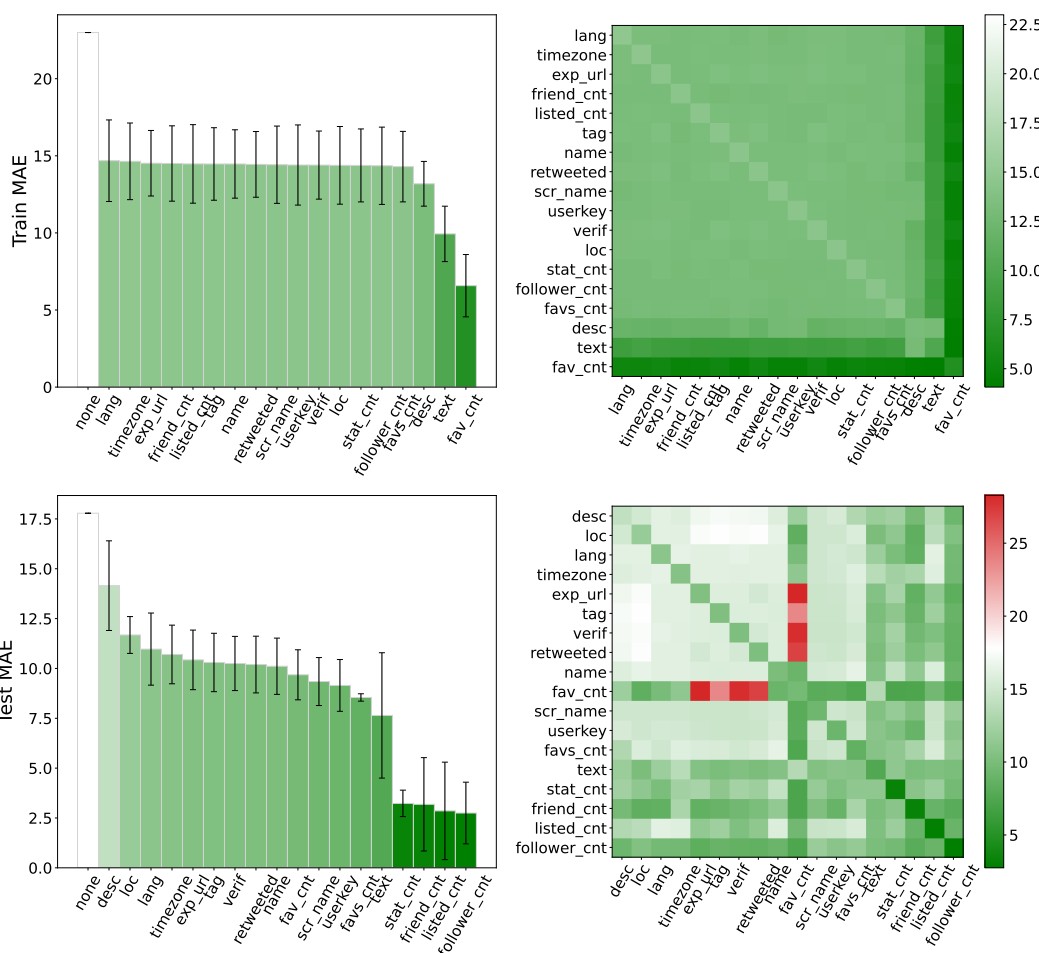

**Figure 12: Neo4j Twitter trolls (node regression, GCN-only results)**. Impact from different properties and their combinations on the MAE. Green: MAE is better than that of a graph with no labels/properties; red: the MAE is worse than that of a graph with no labels/properties.

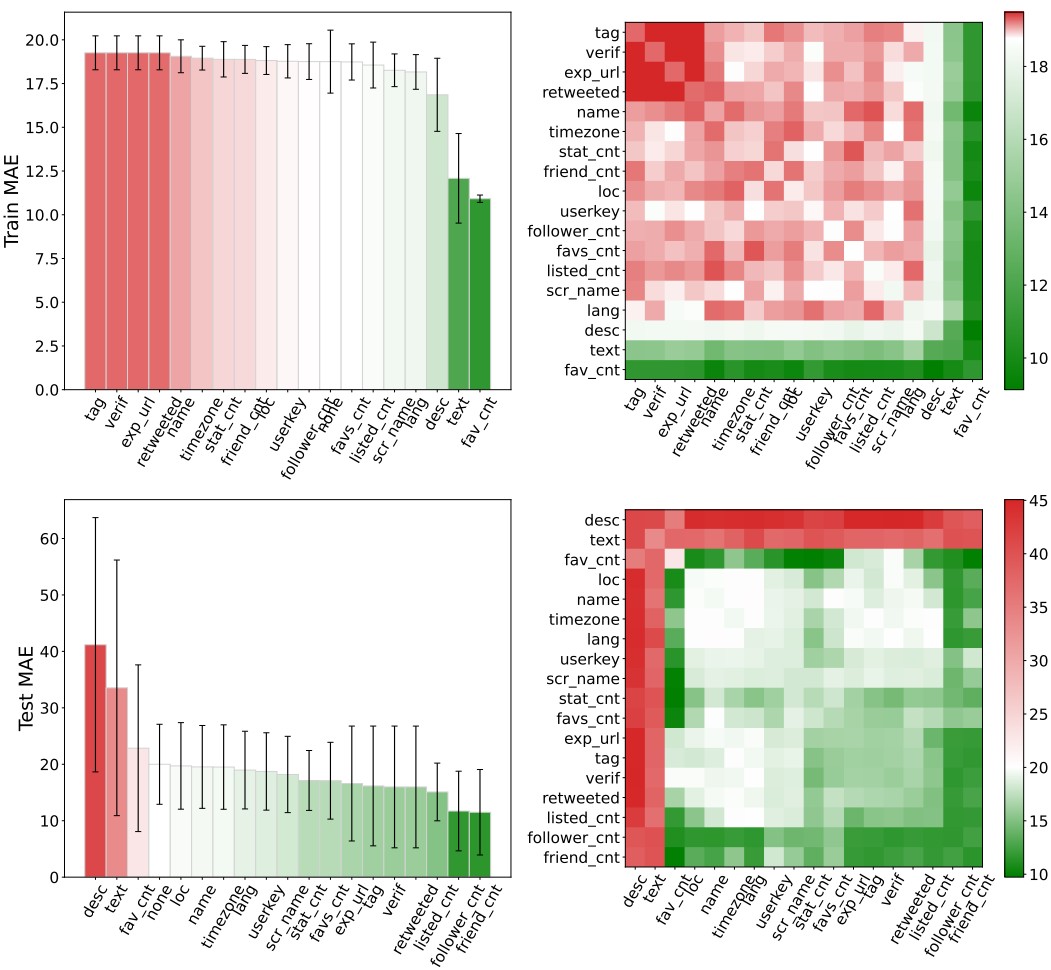

**Figure 13: Neo4j Twitter trolls (node regression, GAT-only results)**. Impact from different properties and their combinations on the MAE. Green: MAE is better than that of a graph with no labels/properties; red: the MAE is worse than that of a graph with no labels/properties.

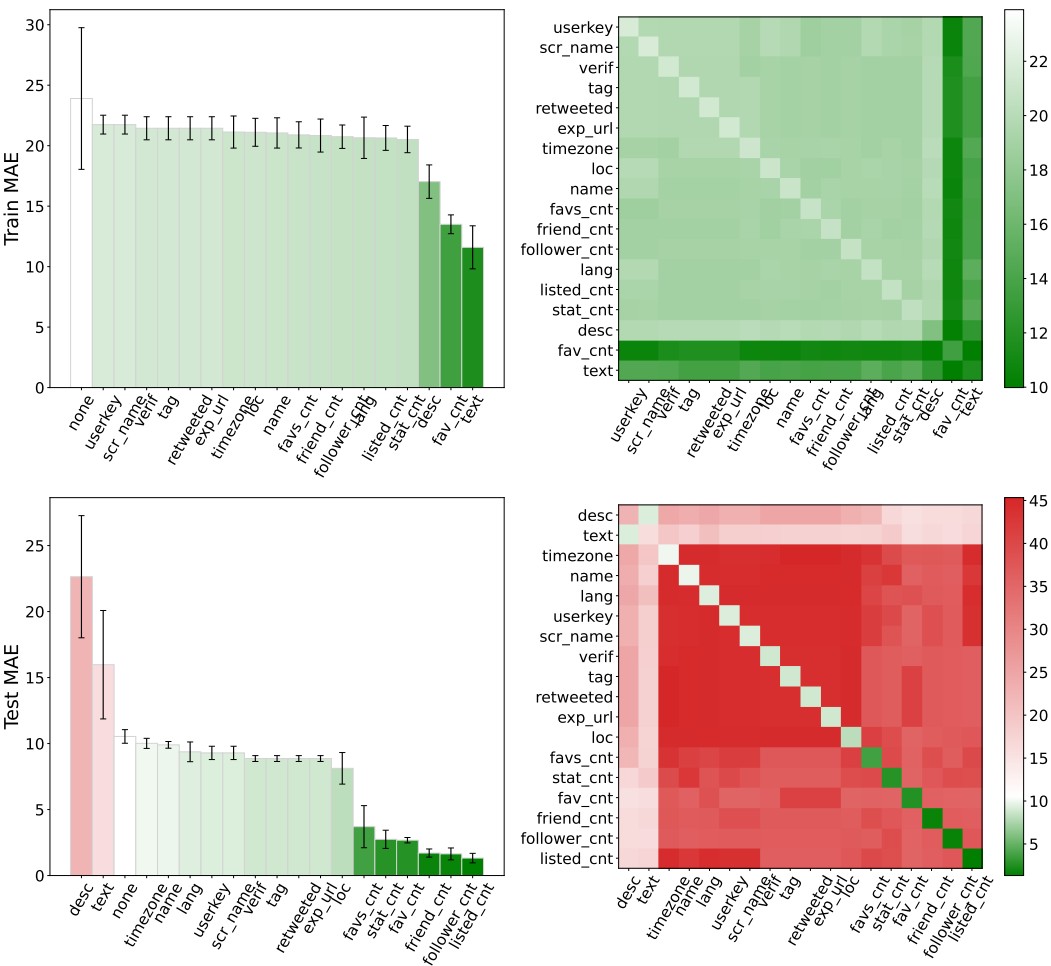

**Figure 14: Neo4j Twitter trolls (node regression, GIN-only results).** Impact from different properties and their combinations on the MAE. Green: MAE is better than that of a graph with no labels/properties; red: the MAE is worse than that of a graph with no labels/properties.

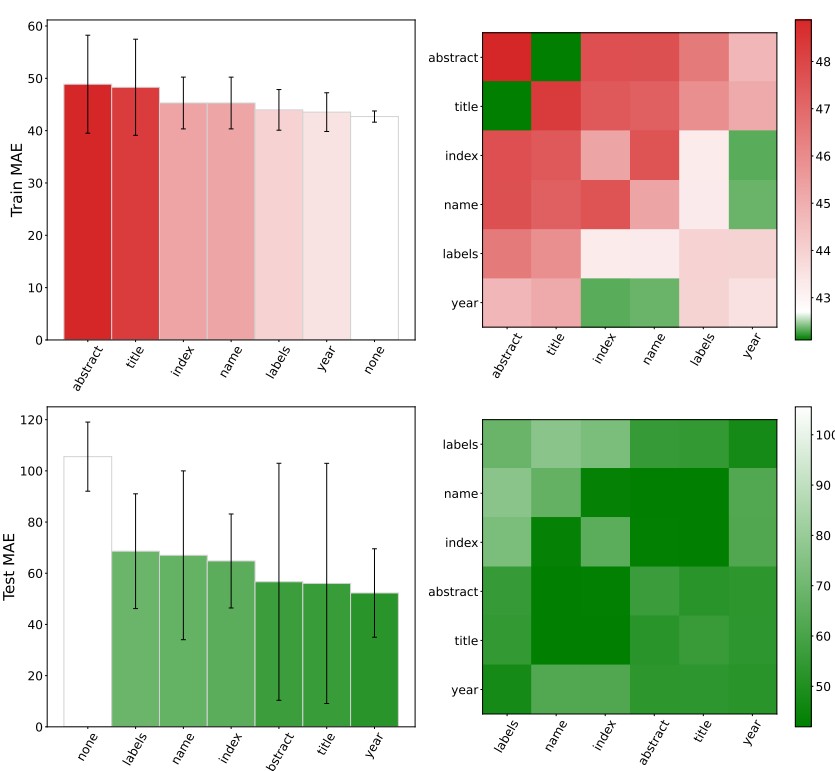

**Figure 15: Neo4j citations (node regression, results aggregated over all three models)**. Impact from different properties and their combinations on the MAE. Green: MAE is better than that of a graph with no labels/properties; red: the MAE is worse than that of a graph with no labels/properties.

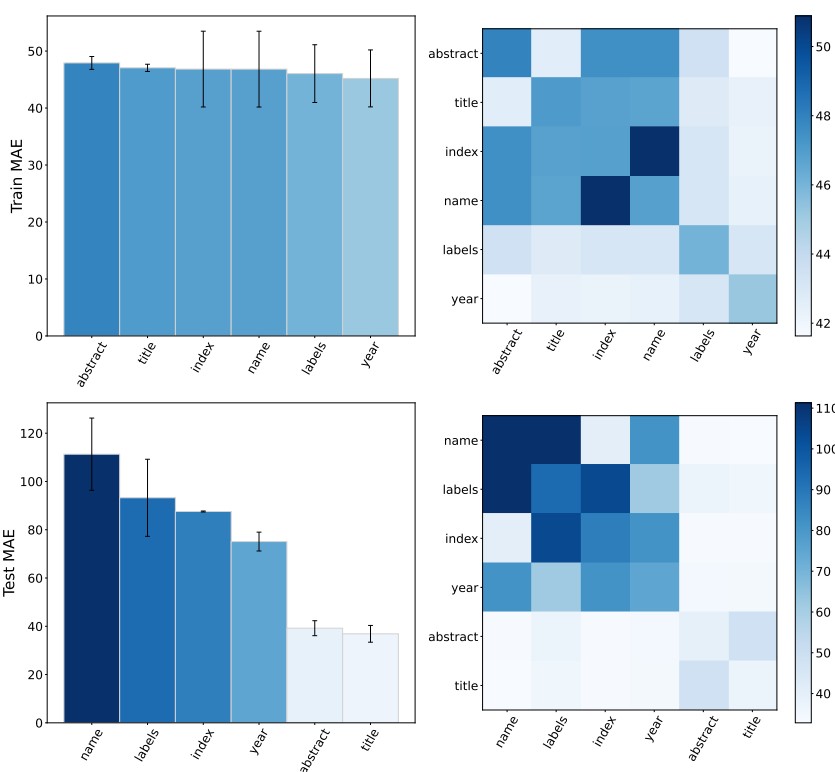

**Figure 16: Neo4j citations (node regression, GCN-only results)**. Impact from different properties and their combinations on the MAE. Here, we do not use green/red colors, because the baselines with no labels/properties could not converge. Instead, we use only one-color (blue) shades to indicate relative improvements.

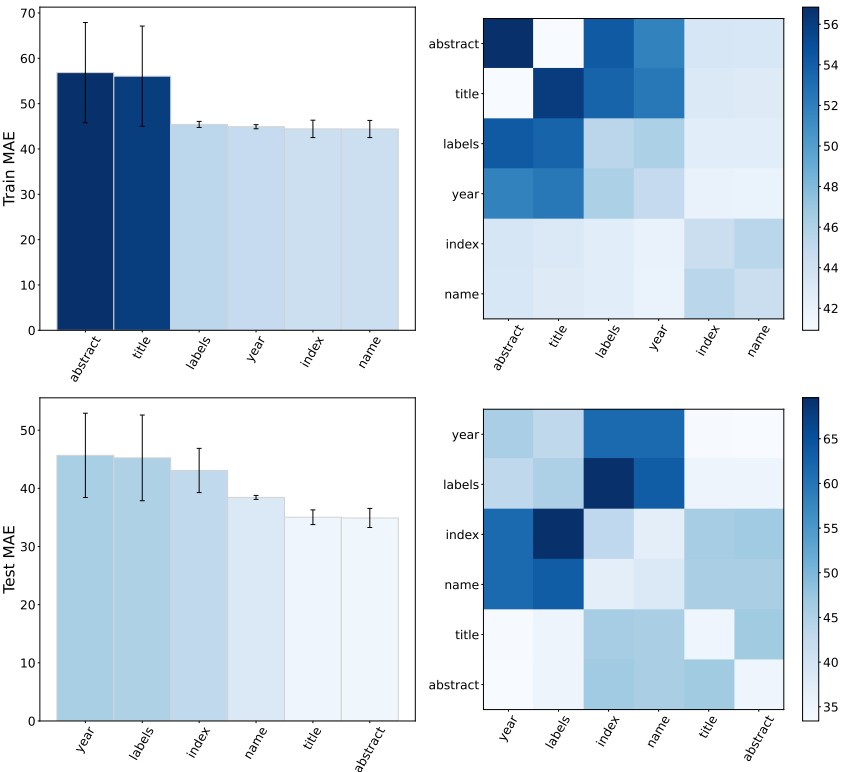

**Figure 17: Neo4j citations (node regression, GAT-only results)**. Impact from different properties and their combinations on the MAE. Here, we do not use green/red colors, because the baselines with no labels/properties could not converge. Instead, we use only one-color (blue) shades to indicate relative improvements.

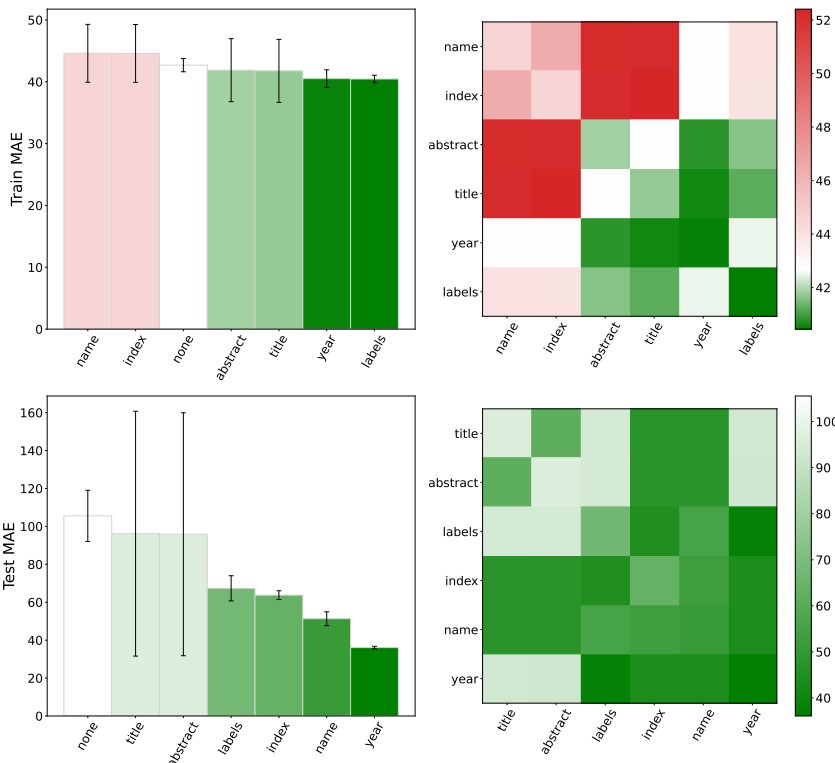

**Figure 18: Neo4j citations (node regression, GIN-only results)**. Impact from different properties and their combinations on the MAE. Green: MAE is better than that of a graph with no labels/properties; red: the MAE is worse than that of a graph with no labels/properties.

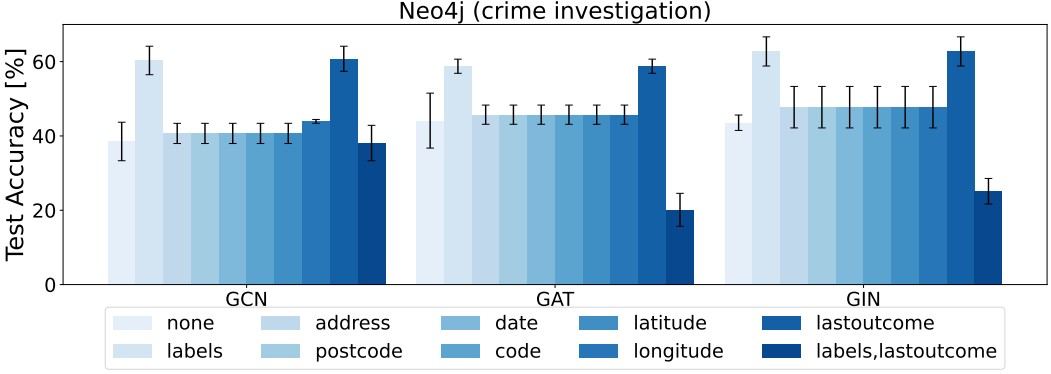

**Figure 19:** Advantages of preserving the information encoded in LPG labels and properties, for node classification in the Neo4j crime investigation dataset.

# C   Details of Embedding Construction

We provide formal specifications of the computed LPG2vec encodings for any vertex $i$ ($\mathbf{x}_i$) and for any edge $(i, j)$ ($\mathbf{x}_{ij}$). The specific fields are as follows: one-hot encoding of the $x$-th label ($l_x$) where $x \in \{1, ..., L\}$, one-hot encoding of the $y$-th property that has $C_y$ potential values ($p_{y,1}, p_{y,2}, ..., p_{y,C_y}$) where $y \in \{1, ..., P\}$, and a string encoding (e.g., BERT) of the $z$-th text feature that has $T_z$ potential fields ($f_{z,1}, f_{z,2}, ..., f_{z,T_z}$) where $z \in \{1, ..., F\}$. This formal description assumes that all the properties are appropriately discretized and - if needed - normalized. The encoding for edges is fully analogous (for simplicity, we assume that the set of labels and properties $L \cup P$ is common for vertices and edges).

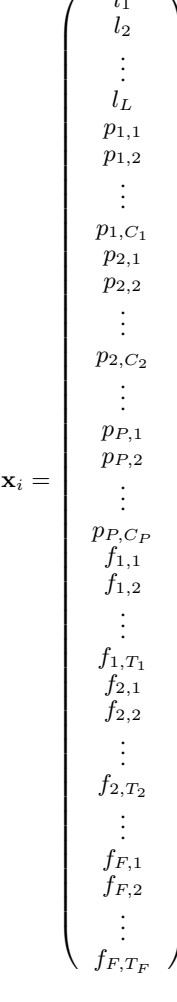

$$\mathbf{e}_{i,j} = \begin{pmatrix} l_1 \\ l_2 \\ \vdots \\ l_L \\ p_{1,1} \\ p_{1,2} \\ \vdots \\ p_{1,C_1} \\ p_{2,1} \\ p_{2,2} \\ \vdots \\ p_{2,C_2} \\ \vdots \\ p_{P,1} \\ p_{P,2} \\ \vdots \\ p_{P,C_P} \\ f_{1,1} \\ f_{1,2} \\ \vdots \\ f_{1,F_1} \\ f_{2,1} \\ f_{2,2} \\ \vdots \\ f_{2,F_2} \\ \vdots \\ f_{F,1} \\ f_{F,2} \\ \vdots \\ f_{F,C_F} \end{pmatrix}$$

# D   Results for Additional Hyperparameters and Models

We also investigate different training split ratios as well as the counts of convolution layers, see Figures 20 and 21. Adding node features generally improves the accuracy across different GNN models and splits. Differences in the training split ratio for the MAKG dataset have little effect on the accuracy. However, in the citations dataset, the accuracy gets worse when it uses more training data. It indicates that, in this dataset and task, the initial 80% split ratio for the training nodes is too high.

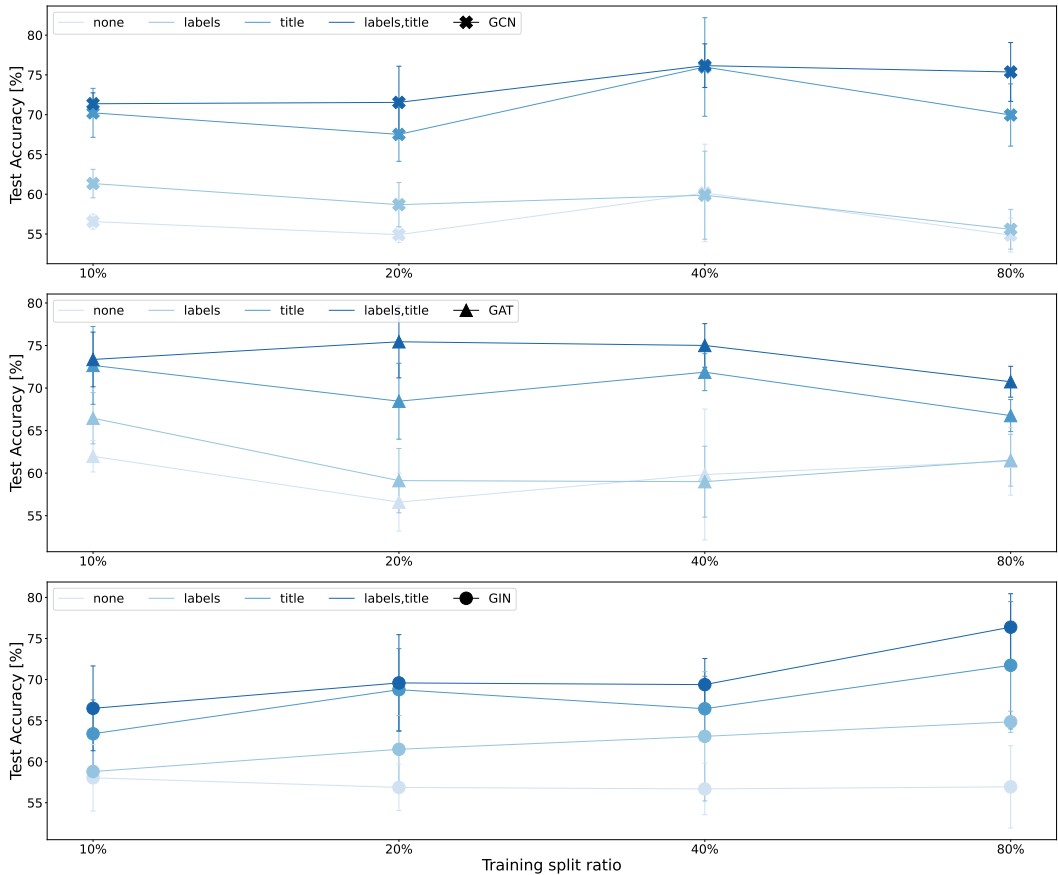

Figure 20: MAKG small (node classification, 4 classes). Impact from different split ratios (the higher the better).

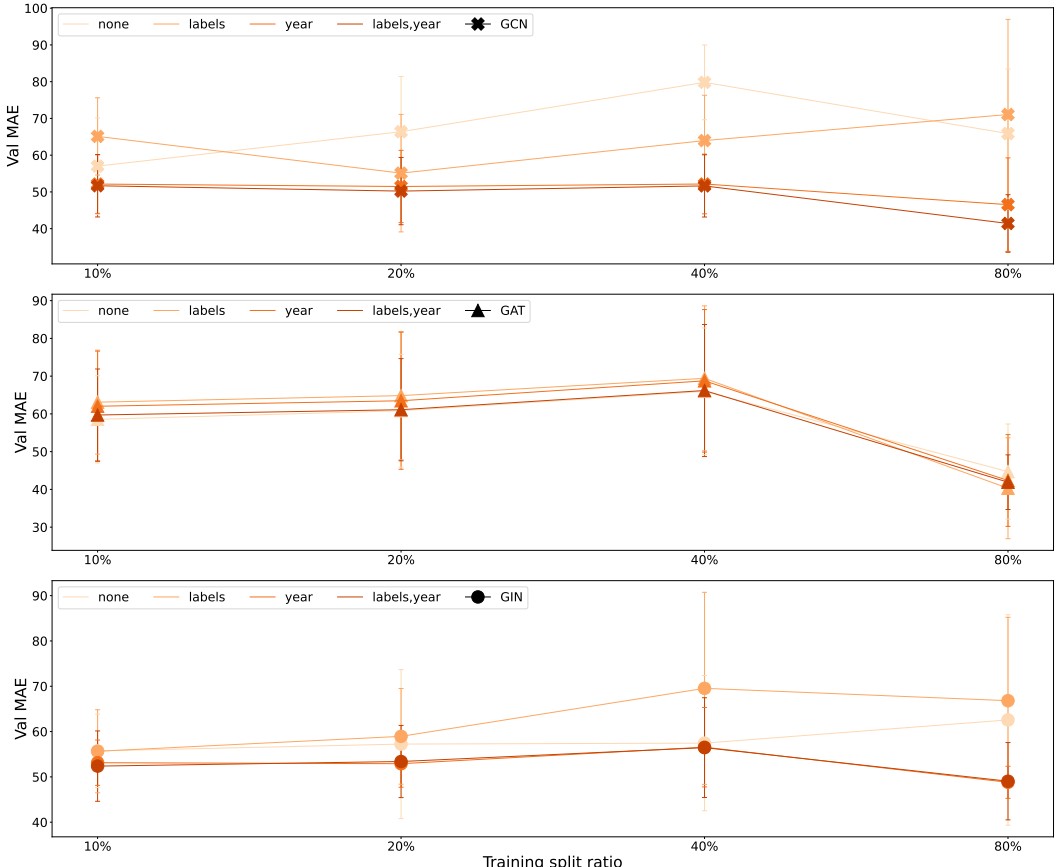

**Figure 21: Neo4j citations (node regression)**. Impact from different split ratios (the lower the better).

We also vary the number convolution layers, see Figure 22. Adding more layers on its own does not bring consistent improvements. This is because the structure of the considered graph datasets usually has a lot of locality and is highly clustered. However, importantly, adding the information from labels and from properties enhances the accuracy consistency across all tried layer counts.

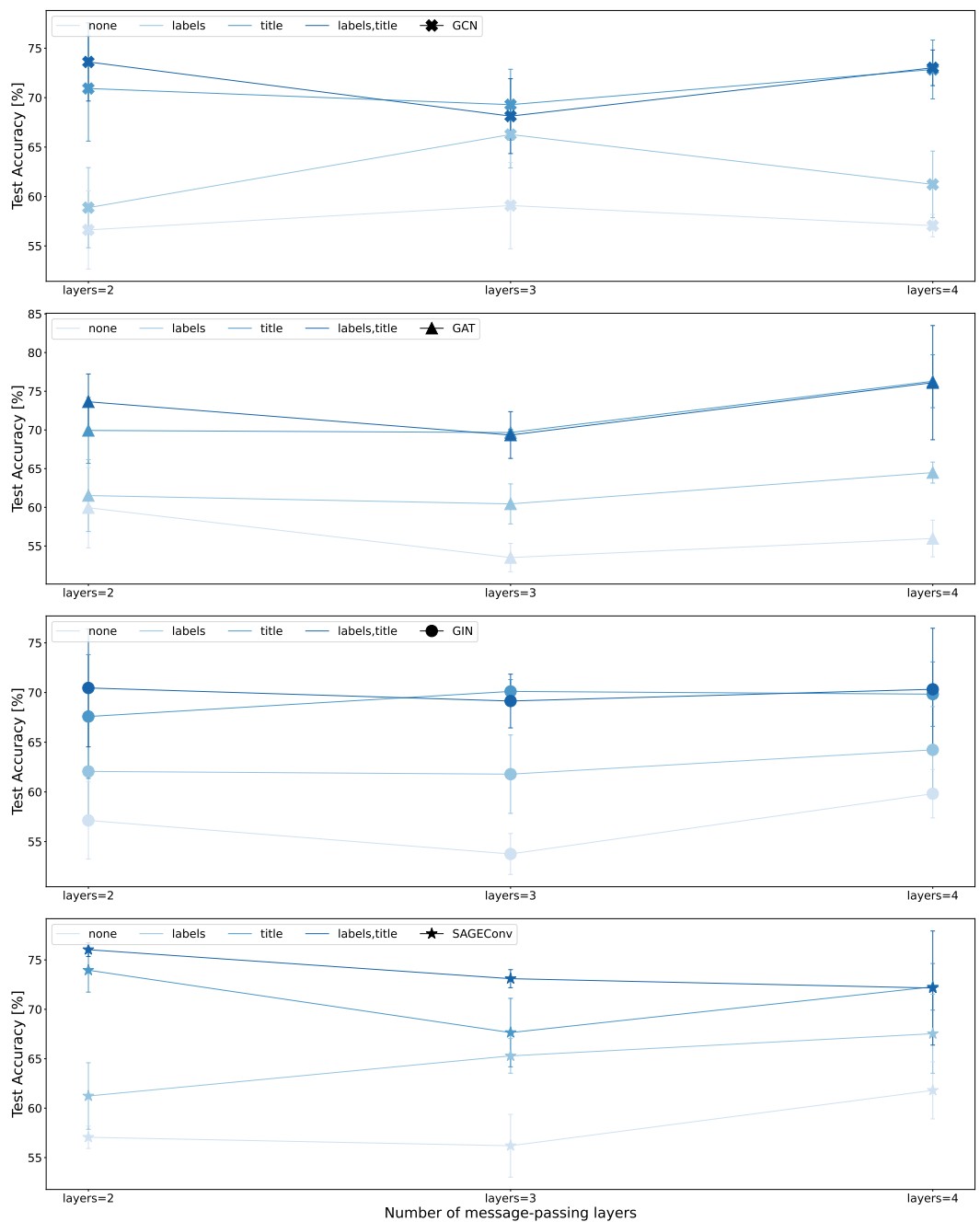

**Figure 22: MAKG small (node classification, 4 classes)**. Impact from different counts of convolution layers (the higher the better).

We also investigated different hyperparameters for LPG2vec embeddings. For example, we experimented with the dimensions of the constructed embeddings. For this, we tried to use an additional MLP to reduce the dimensions of the high dimensional LPG2vec feature vectors, while keeping the information within the features intact. We use two linear layers combined with a dropout layer and the Leaky Relu activation. The dimensions were reduced by different rations, between 20 and $5\times$. This approach on one hand resulted in much smaller input feature vectors, which could visibly reduce the memory storage overheads for particularly large graphs. However, we also observed consistent accuracy losses across all tried datasets and GNN models. We left more extensive experiments into this direction for future work.

Finally, we also investigate additional models, GraphSAGE (Figure 23) and plain MLP (Figure 24). As with GCN, GIN, and GAT, adding more labels and more properties enhances the accuracy. MLP comes with much lower accuracy than GraphSAGE for most tried settings (i.e., with most of labels and properties tried). However, interestingly, it becomes only slightly less powerful than GraphSAGE when including the title property. This further shows the importance of harnessing LPG data - when the right data is included into the initial embeddings, it may offer very high accuracy even without considering the graph structure.

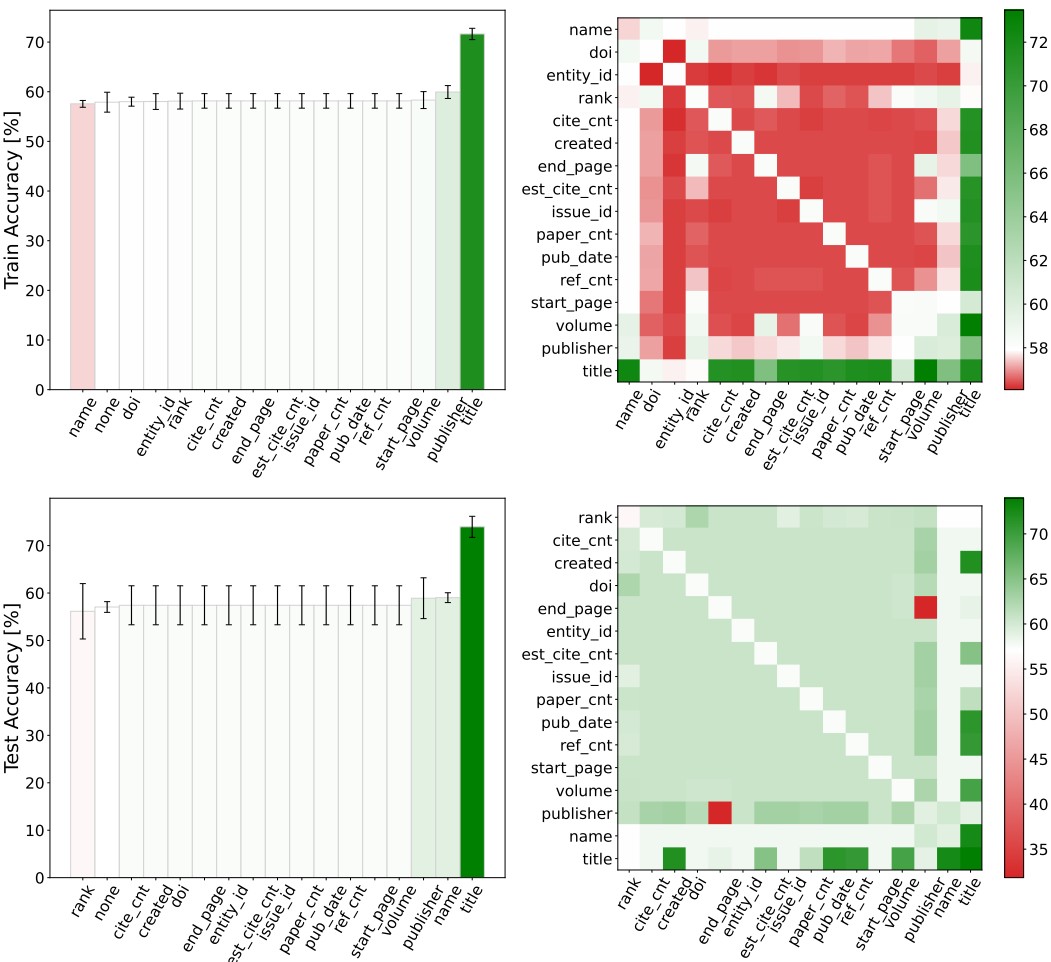

**Figure 23: MAKG small (node classification, 4 classes, GraphSAGE-only results).** Impact from different properties and their combinations on the accuracy. Green: the accuracy is better than that of a graph with no labels/properties; red: the accuracy is worse than that of a graph with no labels/properties.

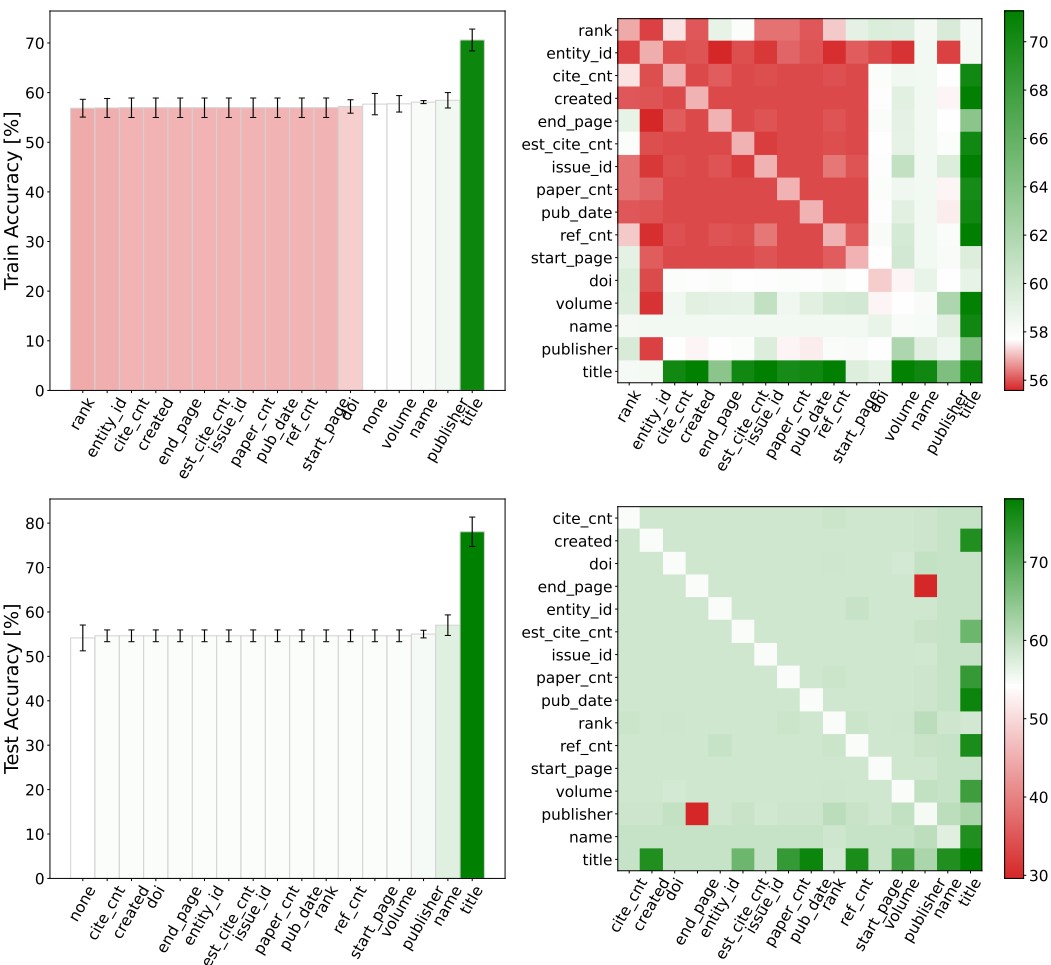

**Figure 24: MAKG small (node classification, 4 classes, MLP-only results)**. Impact from different properties and their combinations on the accuracy. Green: the accuracy is better than that of a graph with no labels/properties; red: the accuracy is worse than that of a graph with no labels/properties.

