# OpenReview forum: "Neural Graph Databases"
_logconference.io/LOG/2022/Conference — LoG 2022 Poster_

### Official Review · Reviewer_x3Ya · 2022-09-29

**Overall Score:** 8
**Confidence:** 3

**Review:**

Contributions of the work
=========================

* The paper introduces LPG2vec, which is basically an encoder that converts LPG labels
and properties into vertex and edge embeddings for subsequent use in graph learning models
such as GCN, GAT, etc.


Strong points
=============

1) The work is novel (modulo inspiration from RDF2vec) and open doors to the improvement
of graph neural networks training.

2) The readability of the paper is quite good. The authors are concise and slowly increase the
complexity of their investigation as more concepts/ideas are introduced.

3) The evaluation, discussion, and analysis of GCN, GAT, and GIN, are extensive and insightful.


Weak points
===========

* No major weak points. I would suggest the authors to possibly include more graph neural nets models in the evaluation, as it would be helpful to verify the behaviour that (sometimes) adding more properties actually hurts the performance of the models.


My recommendation
=================

* The paper is solid, well-written, and it presents an innovative idea with strong potential to impact
graph neural networks training and deployment. Therefore, I'm happy to recommend the paper for acceptance.


Questions authors
=================

* In the "Additional Results" section, is there anything else (in addition to supporting evidence for the
discussion laid out in the main manuscript) we can conclude from these additional experiments?
If so, I would recommend writing those conclusions out.

Additional feedback
===================

* I would think most readers don't appreaciate when there's a substantial amount of text
within a Figure. For example, in Figure 1, the authors wrote two paragraphs within the Figure, ie,
those starting with "A triple models" and "Ensuring fast access". I don't think that's a great idea.
The font is extremely small and because there's not much space the text is squeezed, which makes it
quite hard to read smoothly. In addition, there's quite a number of repetitive elements in Figure 1,
for example the boxes containing "Subject", "Predicate", "Object", "Vertex", etc. Do you really need
six of those "Subject" boxes to illustrate your point? Maybe not. So, I suggest to clean up Figure 1.

---

### Official Review · Reviewer_n77k · 2022-10-17

**Overall Score:** 8
**Confidence:** 3

**Review:**

##########################################################################

Summary:

This paper provides a new method LPGvec to leverage the rich information contained in LPG databases (like label and properties on vertices) to improve performance of many common GNNs (like GCN, GIN, GAT). The main idea is to convert labels and properties on nodes or edges from LPG database into vectors, and concatenate all the vectors for each node or edge.

##########################################################################

Reasons for score:

Overall, I vote for weak accept. The proposed method is simple and efficient which makes it a good candidate for practical applications. My major concerns are about the formal presentation of the approach and the extensiveness of the experiments (see cons below). Hopefully the authors can address my concern in the rebuttal period.

##########################################################################

Pros:

1. The paper is very pleasant to read with helpful figures.
2. The method is simple which is an important advantage for practical use.
3. The method leads to important empirical improvements on the considered datasets

##########################################################################

Cons:

1. Model description (sec. 3.2): It is important to provide the mathematical formulation of the conversion of properties and labels into vectors (i.e. the embedding construction) since this is the key technical trick of the paper. Formal notations allows to remove uncertainty on the conversion interpretation (e.g. how are vectors normalized ?). Right now, all the introduced mathematical in Sec. 2 are not really use to present the model formally in Sec. 3.2 . Action suggestion: Formalize the construction of the embeddings (at least in the appendix).
2. It is not clear if the enriched graph datasets will be made publicly accessible.
3. The experimental setup should be enriched to validate the practicality for the method. What is the impact of the embedding construction type (e.g. normalization vs discretization for continuous scalar, dimension of BERT embedding) ? I would also be interested to see the sensitivity of the method against the split ratio. 80% training nodes might be very high for some applications. Finally, the results would be significantly more convincing with more GNNs architectures like APPNP, GRAND, GPR-GNN or even MLP. Action suggestion: Compare results for different embedding construction. Run experiments with different split ratio including ratio with few training nodes.
4. One important possible confusion is for the term label. In particular, I was sometimes confused if it was referring to class labels which are the target of the downstream task, or the database labels which are one of the additional information of LPG. E.g. l.211 was confusing to me. Action suggestions: Use different words for these two use-cases.
5. It is unclear what is the loss for regression. What do you use for the loss for regression ? It says CE on line l.239 but it would only work for classification.
6. It is interesting to remark that GIN does not work as good as the other GNNs. Do you have an explanation why GIN does not work as good as the others ? It is hard to know what GNNs types can be well combined with LPG2vec. This motivates experimentation with more GNN architectures.
I am happy to improve my score if a majority of the above points are addresses (e.g. with the action suggestions).

---

### Official Review · Reviewer_bJap · 2022-10-21

**Overall Score:** 3
**Confidence:** 4

**Review:**

### Paper Summary
This work presents LPG2vec, a method to transform an arbitrary LPG dataset used by many graph databases into a data structure that can be readily applied to a wide range of graph-representation learning NN structures like GCN, GAT, GIN, etc. Experiments have demonstrated that GNN learning on graphs enhanced with LPG2vec can achieve an up-to-34% increase in performance.

### Strengths
* The paper is well written and well-illustrated
* I was able to run the code artifacts quite easily, which is also well-documented
* The authors have run extensive experiments with well laid-out details on hyperparameters and settings.

### Weaknesses
* The name of this paper is very misleading. “Neural Graph Databases” either implies that the manuscript describes a DBMS that leverages neural network capabilities (aka ML for DB), or that DBMS-like methodology is used to improve a particular neural network architecture (aka DB for ML). This manuscript instead describes a very generic pre-processing / feature engineering technique that converts datasets in a certain data model used in GDBs to a form that is indigestible by GNNs.
* The format and approach of the paper do not seem to be interesting for the audience of LoG. LPG2vec, as it is currently presented, is analogous to ubiquitous techniques used in a variety of existing methodology papers that have an applicative or experimental section. One-hot encoding and sentence transformers are common practices and, therefore, not considered as original in a research setting. These empirical findings, though, may be interesting to a different audience that cares more about concrete implementations.
* I find it very hard to concur with the claim that LPG2vec is the main cause of the 34% increase in performance, as the baseline of such is simply GNN trained without access to certain edge- and node-level labels. It is easily arguable that GNN performed worse not because of the absence of LPG2vec but simply because it does not have access to this information. Figure 5 and 6 should be interpreted as an analysis of the inherent properties of the GNNs and the datasets, not LPG2vec. A more reasonable comparison would be to compare LPG2vec to other encoding methods that made the same amount of information available to GNNs, which, on closer inspection, also highlights the lack of novel contributions.
* The paper, though well written, has casual wordings in lots of places and non-standard terminology that weakens the overall delivery (such as line 59-61, line 222-227, the title of 4.3, etc.)
* From inspection of the code artifacts, it is unclear why Neo4j is emphasized throughout the main text, as the data already comes in a CSV form. There is very little connection to any properties of a graph DBMS besides the usage of the data model LPG, which realistically is already transformed (as in, already extracted from a running Neo4j instance) opaquely before the use of this submitted codebase.


### Recommendation
This paper is a clear rejection as I don’t see the approach taken here is suitable for presentation at the conference.

### Questions
I don’t have any major questions about the paper, as it is easy to follow and clearly written.

### Additional Feedbacks
* Personally, I see a very different interpretation of “marrying graph databases and graph neural networks” that deserves its own paper and is significant and interesting to the conference, in which the training and inference of neural networks take place within the graph database itself. The artifact would be a plugin to one of the existing graph database implementations (e.g., Neo4j) or the fork of an existing project with added functionalities. The challenge here is how to enable efficient inference of graph neural networks (characterized by highly data-parallel implementations on specialized hardware like GPU/NPU) while maintaining the normal behavior of a database: atomicity, consistency, isolation, and durability. It will be conceivably hard to design a system that harnesses GNN’s expressive capabilities like semi-supervised node and edge classification (could be translated into equivalent verbiages from a graph-database perspective), which requires particular designs in memory structure and parallel programming primitives while allowing concurrent, high-throughput and transactional updates like insertion and deletion of node and properties (just like you normally would to a database), which involves an entirely different set of system design considerations such as OS-bypass. It will be an exciting analysis to discover the exact tradeoff curve.

* Minor misprints including the spacing on abstract and usage of text in LaTeX formulae.
* Minor misprints in the code artifact: line 11 of README.md
* Placement of Figure 2 makes 2.1 a bit hard to read.
* Additionally, Figure 7-16 may not be friendly to a black-and-white only / color-blind viewing and printing environment.

### Paper Type
The paper type is correct.

---

### Official Review · Reviewer_Cmds · 2022-10-27

**Overall Score:** 5
**Confidence:** 3

**Review:**

### 1. Contribution

The contribution of this work is to convert graph databases into a vector representation that can be learned by graph neural networks, and to explore which architecture and featurization works best.

### 2. Strong / weak points.

**Strong**
- Graph databases have never been explored before, so there is novelty
- Paper is well written and figures are visually appealing and informative
- This can be a first baseline on which other work build on, and can be a good "dataset paper"

**Weak**
- The work seems trivial on many aspects
- Newer GNN architectures were not tested, including MPNN, PNA, GSN for the message passing, or Graphormer + GraphGPS for Transformers
- Recent progress in positional and structural encodings weren't tested, and they could provide some insight about what kind of information is relevant.
- It would be nice if the proposed graph databases were compared to other datasets like Cora, CiteSeer, Protein, etc. to show how they differ from them.
- Missing non-GNN baselines?

### 3. Recommendation
I vote for a weak reject (5). Confidence 4.

### 4. Arguments

The work can be relevant but seems very superficial. It feels that most of the work is trivial, and this could be framed as a dataset paper instead. Many more modern GNNs are missing, alongside progress in the field of positional encodings. Perhaps they would perform badly due to the kind of tasks that we're facing, but we can't really know without at least a few tests.

However, the work is still relevant to the community, so I'm not super confident in rejecting it.

Willing to upgrade my score.

### 5. Questions

- GCN, GAT, and GIN do not use edge features. However, it seems that edge features are an important component of graph databases. Why stick to these models? Why not try message-passing? How do you incorporate edge info in your current model?
- You only use 2 GNN layers in your models. Why? Do they suffer from over-smooting?  Will adding more layers improve the results? Or is the information very local?
- How important is the local structure and positions of nodes in the graphs? Would doing a random-walk structural encoding [LSPE](https://github.com/vijaydwivedi75/gnn-lspe) help in the performance? What about low-frequency Laplacian eigenvectors? Can you compute them on such large graphs?
- I'm not very familiar with graph databases, but aren't there other methods that you can compare to GNNs? Traditional graph processing methods that you can use as a baseline? What about a simple MLP without the node propagation

### 7. Formatting
Formatting looks good. Paper is well written, figures are nice.

---

### Meta-Review · Area_Chair_cXba · 2022-11-15

**Confidence:** 4
**Recommendation:** Accept

**Meta Review:**

This paper proposes LPG2vec, an approach which transforms an arbitrary labeled property graph (LPG) dataset as is common in industrial graph DBs, into a representation that can be used with most GNN methods.  The work generally aims at making GNNs accessible for GDBs.

- Reviewer Cmds believes the work has some limited novelty, and might be better assessed as a dataset paper or experimental study.  I believe some of this assessment is due to an evaluation of this work as a more traditional academic research contribution rather than an accessibility to research / methods contribution.

- Reviewer bJap raises issue with the scoping of the paper, e.g. the title of "Neural Graph Databases" suggesting something fairly different than what was delivered.  They also take issue with some of the claims about LPG2vec and the reason for its good performance.  Ultimately, this reviewer assessed the work as a high-quality experimental study, but as the authors point out, graph / geometric ML infrastructure-related contributions are aligned with LoG's intended scope.

- Reviewer n77k assessed the work positively, and appreciated the simplicity associated with practical use, but raised a few concerns on formalisms and details.  Most of these are clarified in the appendix by the authors, and the reviewer's assessment is made more positive.

- Reviewer x3Ya has a similar assessment as n77k and asks for a few additional results, which the authors provide.

*Strengths*

- S1: The idea proposed is simple, and is shown to work well.
- S2: The authors' proposal increases the accessibility of graph learning methods to a broader (GDB) community.

*Weaknesses*

- W1: The work has limited technical novelty given its design is fairly straightforward.
- W2: Some details about formalism and clarity is not clear from the main paper text (e.g. in Section 3).

Overall, S2 > S1 > W2 > W1 leads to my positive recommendation about this work.  Since accessibility-related improvements based on infrastructure / deployment considerations are impactful and explicitly within the LoG CFP, I believe this work has a place at the conference.

---

### Decision · Program_Chairs · 2022-11-23

Accept (Poster)